# A neuronal ensemble encoding adaptive choice during sensory conflict in *Drosophila*

Preeti F. Sareen [1], Li Yan McCurdy [1,2] & Michael N. Nitabach [1,3,4 ✉]

Feeding decisions are fundamental to survival, and decision making is often disrupted in disease. Here, we show that neural activity in a small population of neurons projecting to the fan-shaped body higher-order central brain region of *Drosophila* represents food choice during sensory conflict. We found that food deprived flies made tradeoffs between appetitive and aversive values of food. We identified an upstream neuropeptidergic and dopaminergic network that relays internal state and other decision-relevant information to a specific subset of fan-shaped body neurons. These neurons were strongly inhibited by the taste of the rejected food choice, suggesting that they encode behavioral food choice. Our findings reveal that fan-shaped body taste responses to food choices are determined not only by taste quality, but also by previous experience (including choice outcome) and hunger state, which are integrated in the fan-shaped body to encode the decision before relay to downstream motor circuits for behavioral implementation.

[1] Department of Cellular and Molecular Physiology, Yale University, New Haven, CT, USA. [2] Interdepartmental Neuroscience Program, Yale University, New Haven, CT, USA. [3] Department of Genetics, Yale University, New Haven, CT, USA. [4] Department of Neuroscience, Yale University, New Haven, CT, USA. ✉email: michael.nitabach@yale.edu

  1

Animals integrate food-related sensory information from their external environment with their internal state in order to make adaptive decisions. Often food-related sensory information is conflicting in valence. For example, *Drosophila* flies forage on decomposing fruits and must balance obtaining essential nutrition with avoiding toxins, pathogens, etc. As flies forage, sweet and bitter taste receptors on their legs and wings signal the presence of sweet nutritive food and bitter potential toxins[1]. Flies must adaptively weigh and integrate this conflicting information before consumption to enhance survival fitness. While some studies have shown integration of sweet and bitter tastes at the sensory neuron level[2–4], very little is known about how taste and other external sensory cues are integrated with internal state of the animal to form a feeding decision in the higher brain. Here, we report an ensemble of neurons in the central brain of a hungry fly that makes value-based decisions, using an experimental paradigm in which freely foraging flies sample and choose between different sweet and bittersweet foods.

## Results

**Hungry flies make tradeoffs when faced with conflicting sensory information.** We quantified the choices of wild-type flies deprived of food for different durations between a range of increasing concentration of sweet (sucrose-only) food option and a constant bittersweet (sucrose + quinine) option, with each option including either a red or blue food dye (Fig. 1a). Food choice was quantified by scoring the color of the food in each fly's abdomen at the end of the assay. A preference index was then

calculated by subtracting the number of flies that ate bittersweet food from the number of flies that ate sweet food divided by the total number of flies that consumed food in that trial. When choosing between a low sucrose concentration sweet option and a high sucrose concentration bittersweet option, flies prefer higher sucrose bittersweet (Fig. 1b), as quantified by color of food ingested. As sucrose concentration of the sweet choice increased, flies increasingly preferred it over bittersweet. This dose-dependent change in preference suggests that when the sweet option reaches sufficient sucrose, the caloric advantage in choosing a less palatable bittersweet food is outweighed by the danger-avoidance advantage of the sweet option (Fig. 1b). In the absence of bitter, flies always chose the sweeter option (Supplementary Fig. 1a), suggesting that there was no saturation of sweet taste sensation for behavioral discrimination even at the highest sucrose concentration tested (500 mM sucrose). At ten- and fivefold sucrose concentration ratios (Fig. 1b, 50 mM vs. 500 mM sucrose + 1 mM quinine, 100 mM sucrose vs. 500 mM + 1 mM quinine at 21 h) almost all flies consumed only a single food option with half of them consuming only sweet and half consuming only bittersweet resulting in equal-preference for sweet and bittersweet at these conditions. The equal-preference point varied with food-deprivation duration, shifting preference from sweet toward bittersweet with increasing food-deprivation duration (Fig. 1b, Supplementary Fig. 1d–f, Supplementary Table 1), suggesting an external taste sensation and internal hunger state equilibrium at the equal-preference concentration ratios. The equal-preference point depends on the sucrose concentration

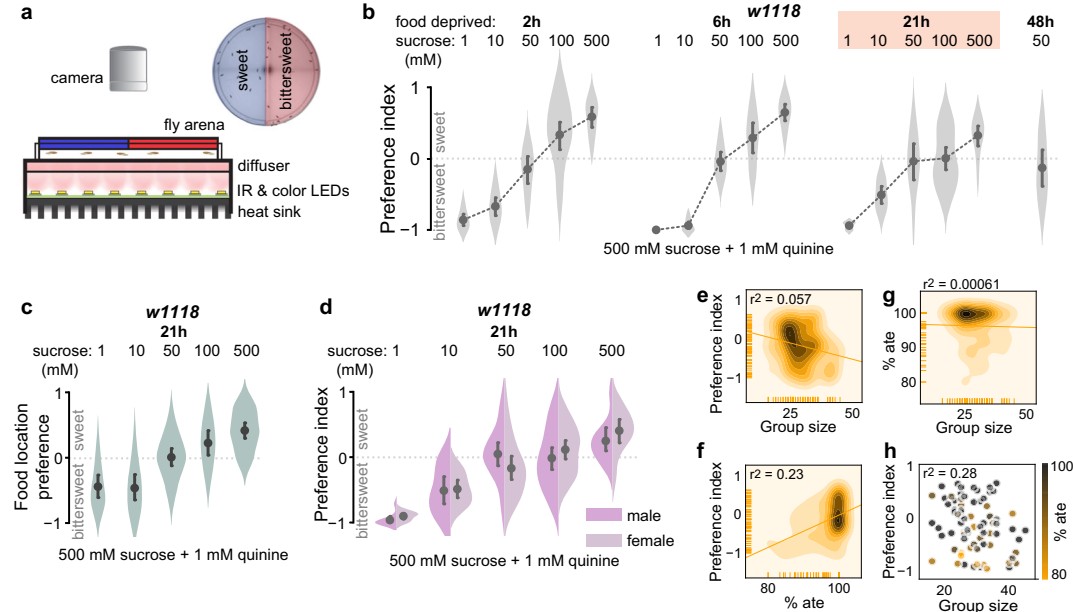

**Fig. 1 Hungry flies make trade-offs between the appetitive and aversive value of food. a** Schematic of the two-choice decision making assay. Sweet and bittersweet foods are prepared in dissolved agarose, mixed with food dyes, and gelled in a circular arena. Flies are introduced into the dark food arena to walk and consume freely for 5 min. Flies are then anaesthetized and abdomen color is recorded, indicating ingested food choice. Preference index is calculated as (no. of sweet color flies + 0.5 purple flies) − (no. of bittersweet color flies + 0.5 purple flies)/total no. of flies that ate, where positive index means flies preferred sweet and negative index means flies preferred bittersweet. **b** Preference index of wild-type (*w1118*) flies deprived of food for increasing durations indicate the tradeoff between the sweet and bittersweet food and reveal equal preference when the choice is between 50 mM sucrose-only and 500 mM sucrose + 1 mM quinine. **c** Position preference index (i.e., sweet or bittersweet sector preference as defined by the location of the flies at the end of the assay) matches ingested food preference. **d** Preferences of male and female flies within each group were indistinguishable for all conditions tested. There was no significant correlation between preference index and group size per trial (**e**), preference index and % of flies that ate per trial (**f**) or % of flies that ate per trial and group size (**g**). **h** Neither group size nor % flies that ate significantly predicted preference index in a multiple regression model, indicating no interaction between these measures. **b–d** Plots show mean ± 95% CI, violins depict kernel density estimation of the underlying data distribution with the width of each violin scaled by the number of observations at that y-value. Each violin summarizes 10 ≤ trials ≤ 30 with mode = 10. **e–h** Heatmaps depict bivariate distribution visualized using a kernel density estimation procedure; darker regions indicate higher data density. $r^2$ is the square of Pearson's coefficient. See Supplementary Table 1 for sample size and statistics. Nonsignificant differences are not depicted in figures.

ratio between the two options and not absolute concentration (Supplementary Fig. 1b). These results indicate that hungry flies tradeoff the appetitive (sweet) and aversive (bitter) values of food in making feeding decisions, and are consistent with known effects of bitter compounds on feeding, in which increasing bitterness of food decreases feeding[4].

To further characterize fly behavior in the decision task and determine whether sensory cues other than taste could affect food choice, we also determined the spatial location of each fly at the end of the decision task. As expected, position preference mirrored ingested food preference (Fig. 1c). We also addressed whether size or sex ratio of groups of flies tested together influenced the decision, since social interactions driven by group size[5,6] and sex ratio can affect fly behavior. There was no effect on ingested food preference of male-to-female ratio within groups of flies tested together (Fig. 1d). At the equal-preference condition (21 h deprivation, 50 mM sucrose vs. 500 mM sucrose + 1 mM quinine), neither food preference and group size (Fig. 1e), food preference and percent of flies that ate (Fig. 1f), nor percent of flies that ate and group size (Fig. 1g) were correlated. There was also no significant prediction of preference index by group size or percent of flies that ate in a multiple regression model (Fig. 1h, Supplementary Table S1), indicating no interaction between these variables in the decision task. These results suggest that taste is the most salient sensory cue affecting food choice in the decision assay.

**A decision making neuronal ensemble converges on the fan-shaped body.** During foraging, animals estimate values of internal state and external sensory parameters, such as hunger level, valence of available foods, location of food, etc. These value estimates are then integrated to form a decision. We hypothesized that value estimates of internal state and external environment encoded by neuromodulatory neurons will be required inputs to higher-order brain regions for integration and generation of decisions. To test this hypothesis, we genetically targeted acute optogenetic activation and inhibition to specific neural subsets in neuromodulatory and higher-order brain centers using the GAL4-UAS binary expression system[7], while flies actively sampled and consumed food at the equal-preference condition (Fig. 1a, b, 21 h food deprivation, 50 mM sucrose vs 500 mM sucrose + 1 mM quinine). For optogenetic activation, we used red-light-sensitive CsChrimson channelrhodopsin[8], and for optogenetic inhibition we used green-light-sensitive anion-conducting channelrhodopsin GtACR1[9]. This optogenetic interrogation of modulatory neurons and higher order brain regions revealed neuropeptidergic neurons (Leucokinin, Allatostatin A, NPF, DH44), subsets of dopaminergic neurons (PPL1-γ2α'1, PPL1-α3, PAM-α1), and a narrow subset of fan-shaped body layer 6 neurons (FBl6) whose activation or inhibition significantly shifted food choice in the equal-preference condition (Fig. 2a).

The specific effects of activation or inhibition of restricted neuron populations on the sweet versus bittersweet decision provides insights into the roles of such populations in value estimation and integration. Optogenetic activation of Leucokinin (Lk) neurons suppressed feeding in food deprived flies (Fig. 2a, b left panel, Supplementary Fig. 2a, b), suggesting that Lk may encode or relay metabolic state information. To confirm that Lk secreted by these neurons was the molecular basis of this feeding suppression, we simultaneously knocked down Lk expression with genetically encoded RNAi while optogenetically activating Lk neurons. While almost no flies consumed food when Lk neurons were activated without Lk RNAi, simultaneous Lk knockdown prevented this feeding suppression (Fig. 2b left panel, Supplementary Fig. 2a, b). This indicates that Lk secretion

mediates feeding suppression by Lk neurons. Optogenetic silencing of Lk neurons shifted preference towards bittersweet (Fig. 2a, b left panel, Supplementary Fig. 2a, b). Feeding suppression induced by Lk neuron activation implies a decrease in perceived hunger of food deprived flies, consistent with the shift towards higher calorie bittersweet food by Lk neuron inhibition reflecting increased perceived hunger. Optogenetic activation of Allatostatin A (AstA) neurons shifted preference towards sweet, while inhibition shifted preference towards bittersweet (Fig. 2a–c left panel). We confirmed that AstA was the molecular basis of this shift in preference by simultaneous AstA RNAi knockdown and optogenetic activation of AstA neurons (Fig. 2c left panel). Optogenetic activation of NPF neurons shifted preference towards sweet, and this shift was abolished by simultaneous activation and NPF RNAi knockdown (Fig. 2a–d left panel). Optogenetic activation of DH44 neurons had no significant effect, but inhibition shifted preference towards bittersweet (Fig. 2a–e left).

Dopaminergic subsets involved in aversive memory and modulation of flight and feeding (Fig. 2a PPL1-γ2α'1)[10,11], taste conditioning (Fig. 2a PPL1-α3)[12], and long-term memory (Fig. 2a PAM-α1)[13] also affected food choice. Specifically, optogenetic activation of PPL1-γ2α'1 and PAM-α1, while optogenetic inhibition of PPL1-α3 subsets shifted preference towards bittersweet (Fig. 2a). Optogenetic activation of various subsets of mushroom body Kenyon cells (KC), a brain region that controls higher-order behaviors and receives projections from the identified dopaminergic neurons, did not affect preference (Fig. 2a). The dopaminergic neurons influencing sweet-bittersweet food choice have projections in several other brain regions in addition to mushroom body, such as, superior medial protocerebrum (SMP), superior lateral protocerebrum (SLP), and superior intermediate protocerebrum (SIP). We thus conclude that dopaminergic projections to these other non-mushroom body brain regions mediate dopaminergic modulation of sweet-bittersweet decision making. Additionally, optogenetic activation of a large population of serotonergic neurons targeted by Trh-GAL4 ceased fly movement and resulted in no feeding (Supplementary Fig. 2a). A previous study has also reported that activating large populations of serotonergic neurons using Trh-GAL4 induces quiescence and inhibits feeding and mating in flies. This study also identified specific neural clusters that suppress mating, but not feeding, suggesting that serotonin does not uniformly act as a global, negative modulator of general arousal[14].

Optogenetic inhibition of a specific subset of FBl6 neurons shifted preference towards bittersweet (Figs. 2a, 3a left panel). We confirmed that FBl6 neurons produced this shift in preference by genetically repressing the expression of GAL4 in FBl6 neurons during optogenetic inhibition. Specifically, we used GAL80 enhancer-trap[15], to express LexAop-GAL80 in FBl6 neurons using 84C10-LexA, while driving UAS-GtACR1 expression using c205-GAL4 active in FBl6 neurons. GAL80 is a yeast repressor that blocks GAL4 activity by binding to its transcriptional activation domain[15]. This targeted manipulation abolished the preference shift induced by driving UAS-GtACR1 in c205-GAL4 in FBl6 neurons, thus confirming the role of FBl6 neurons in producing preference shift upon optogenetic inhibition (Fig. 3a left panel, "84C10 > GAL80 + c205>Gt", genotype: 84C10-LexA; c205-GAL4 > LexAop-GAL80;UAS-GtACR1). Value estimates of internal state and external sensory cues, which are likely computed by identified modulatory neurons, are crucial for decision making. An animal's internal state is produced by the combination of its current sensory and internal environment, such as hunger level, thirst level, motivation to feed, sleep or mate etc. To facilitate description and interpretation of our results in this study we define "encoding" and "integration" as follows. By

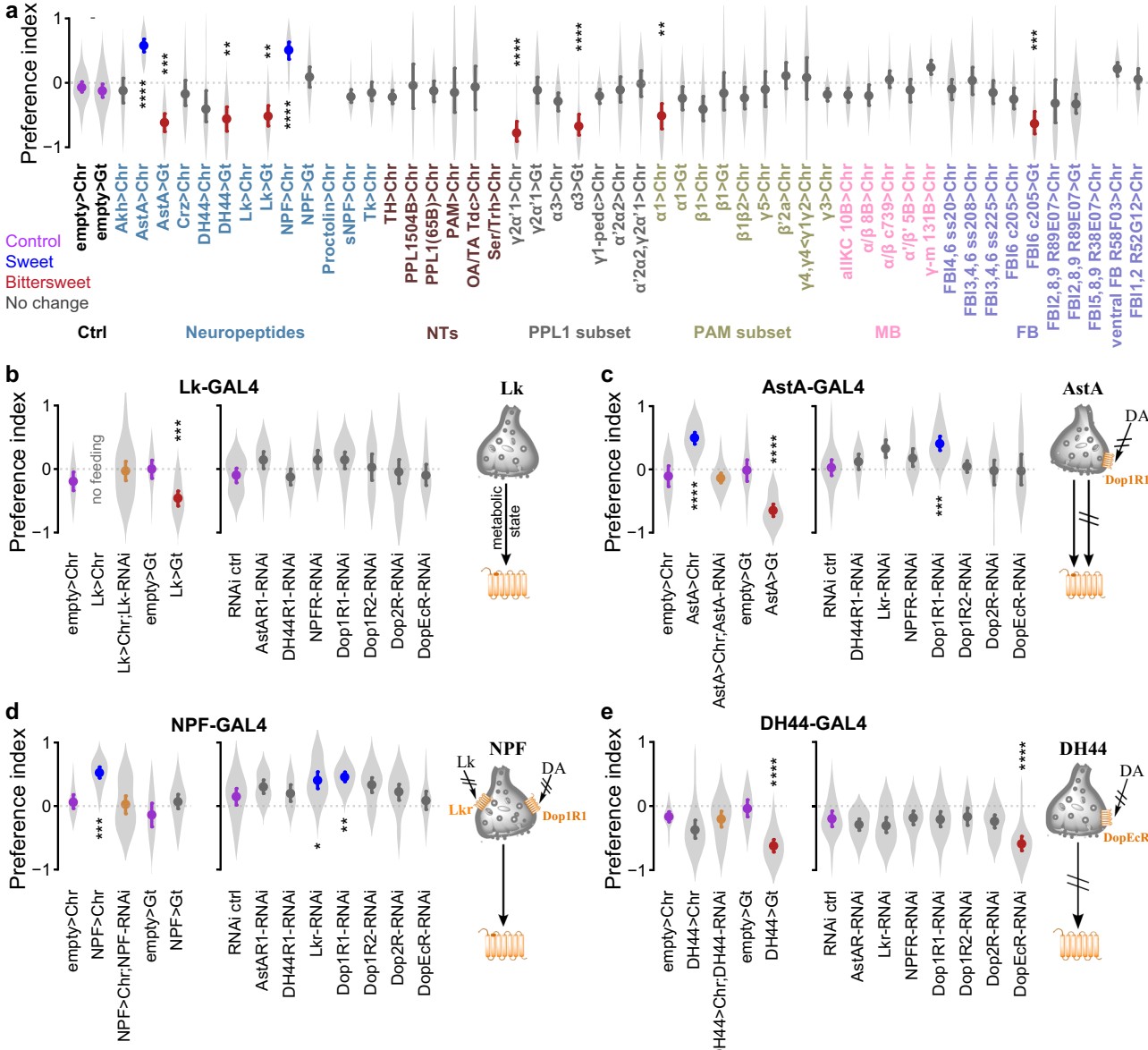

**Fig. 2 A decision making neuronal ensemble revealed by combined optogenetic and chemoconnectomic strategy. a** Cell-specific optogenetic activation and inhibition screen was performed at the equal preference condition (50 mM sucrose vs. 500 mM sucrose + 1 mM quinine) using flies deprived of food for 21 h. Neuronal subsets were genetically targeted using the GAL4-UAS binary expression system. Red-light-sensitive CsChrimson (Chr) was used for optogenetic activation and green-light-sensitive GtACR1 (Gt) for optogenetic silencing. Optogenetic manipulation of AstA, DH44, Lk, and NPF neuropeptides; PPL1 γ2α′1, PPL1 α3, and PAM α1 dopaminergic subsets; and fan-shaped body layer 6 neurons (FBl6) significantly shifted preference away from equal preference compared to respective empty>Chr and empty>Gt controls. Purple mean point = control, blue mean point = significant sweet preference, red mean point = significant bittersweet preference, yellow mean point = simultaneous optogenetic activation and RNAi. **b** (left) Optogenetic activation of Leucokinin (Lk) neurons suppressed feeding in food deprived flies, while optogenetic inhibition shifted the preference towards bittersweet. Simultaneous Lk RNAi in Lk neurons abolished the suppression of feeding induced by optogenetic activation. **b** (right) RNAi in Lk neurons of analogous receptors for other candidate neuromodulators had no effect. **c** (left) Optogenetic activation of Allatostatin A (AstA) neurons shifted preference towards sweet while optogenetic inhibition shifted it towards bittersweet. Simultaneous AstA RNAi abolished the preference shift induced by optogenetic activation. **c** (right) Dop1R1 RNAi in AstA neurons shifted preference towards sweet. **d** (left) Optogenetic activation of NPF neurons shifted preference towards sweet. Simultaneous NPF RNAi abolished the preference shift induced by optogenetic activation. **d** (right) Lkr or Dop1R1 RNAi in NPF neurons each shifted preference towards sweet. **e** (left) Optogenetic activation of DH44 neurons had no effect on preference, while optogenetic inhibition shifted preference toward bittersweet. **e** (right) DopEcR RNAi in DH44 neurons shifted preference towards bittersweet. Schematics summarize the decision relevant information flow for specific neuropeptides. Plots show mean ± 95% CI, with violins depicting full data distribution; 5 ≤ trials ≤ 30 per violin, mode = 10. See Supplementary Table 1 for sample size and statistics. $p < 0.00001 = ****$, $p < 0.0001 = ***$, $p < 0.01 = **$, $p < 0.05 = *$.

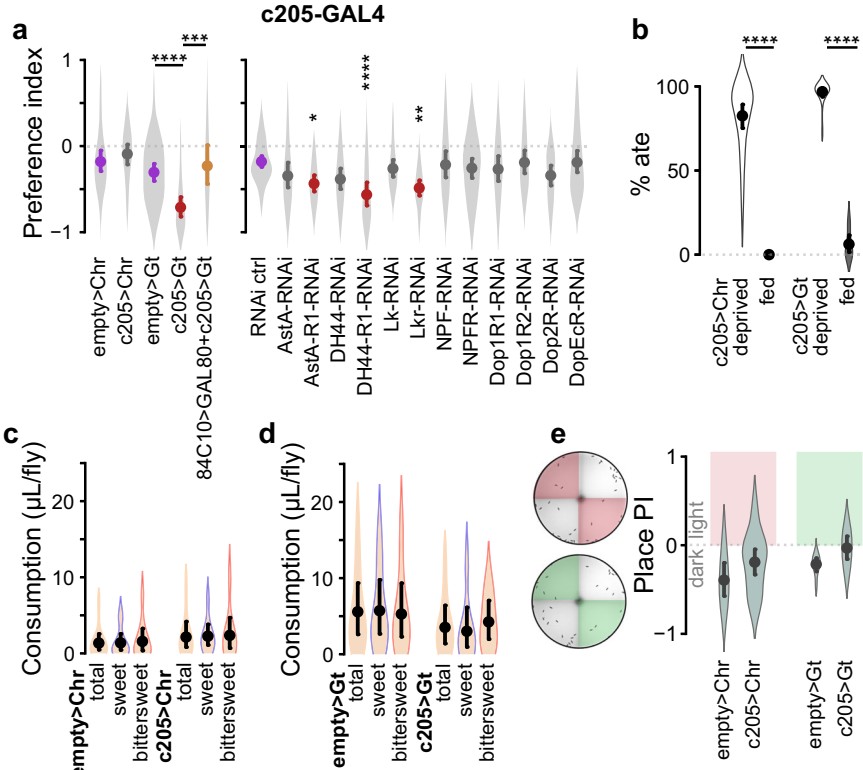

**Fig. 3 Fan-shaped Body layer 6 is the convergence node of a decision making ensemble. a** (left) Optogenetic activation of FBl6 neurons did not affect preference. In contrast, optogenetic inhibition shifted preference towards bittersweet. Simultaneous repression of GAL4 expression in FBl6 neurons using GAL80 during optogenetic inhibition abolished the preference shift induced by optogenetic inhibition. This manipulation confirmed the role of FBl6 neurons in producing preference shift upon optogenetic inhibition. **a** (right) RNAi knockdown of AstA-R1, DH44-R1, or Lkr receptors in FBl6 shifted preference toward bittersweet. **b** Optogenetic activation or inhibition of FBl6 did not induce feeding in fed flies or prevent feeding in food-deprived flies. Optogenetic activation (**c**) or inhibition (**d**) of FBl6 did not change total food consumption per fly compared to respective empty-GAL4 controls. Food consumption was quantified by extracting red and blue food dyes from flies after the assay and measuring the extract absorbance using UV–vis spectrometry. Concentrations of individual dyes were calculated per trial by interpolating absorbance in the standard curve for each dye. Total volume of agarose ingested was then calculated from red and blue dye concentrations and divided by number of flies that ate red and blue food in that trial. Total food volume is the average of consumption by each fly regardless of type of food consumed. Within each group, there was no significant difference in sweet vs. bittersweet food consumption per fly. Optogenetic inhibition of FBl6 shifted preference towards bittersweet (**a**, left), because more flies consumed bittersweet than sweet. **e** Optogenetic activation or inhibition of FBl6 did not significantly change preference (place PI) of flies for illuminated vs. non-illuminated sectors of the arena, indicating that neither activation nor inhibition of FBl6 is inherently rewarding or punishing. Plots show mean ± 95% CI, with violins depicting full data distribution. Statistically different means are shown in different color as in Fig. 2. See Supplementary Table 1 for sample size and statistics. $p < 0.00001 =$ ****, $p < 0.0001 =$ ***, $p < 0.01 =$ **, $p < 0.05 =$ *.

"encoding" of hunger state we mean the estimation of hunger level by specific neurons. By "integration" we mean combining of hunger state with other internal state and external sensory information. Because many of the neuromodulators identified in our screen and their receptors[16–21] co-localize in the fan-shaped body, we hypothesize that neuromodulatory inputs encoding value estimates are integrated in FBl6, where a decision is generated.

To test this hypothesis and determine whether the neurons we identified in our optogenetic screen are connected in a behaviorally relevant ensemble, we employed a chemoconnectomics approach[22] using cell-specific genetically encoded RNAi knockdown of neuropeptide and dopamine receptors. By knocking down a specific receptor in a candidate target neuron, we test whether that neuron receives direct input from a neuron secreting the cognate ligand for that receptor. Knockdown of neuropeptide or dopamine receptors in Lk neurons did not shift preference (Fig. 2b right panel). This indicates that Lk neurons do not receive direct inputs from decision-relevant dopaminergic neurons or from neurons secreting the ligands for tested neuropeptide receptors, and implies that Lk neurons must receive

food preference and hunger related information indirectly from other neurons. RNAi knockdown of Dop1R1 dopamine receptor in AstA neurons shifted preference towards sweet (Fig. 2c right panel), indicating that AstA neurons receive food-preference-relevant dopaminergic inputs. RNAi knockdown of Lkr and Dop1R1 in NPF neurons shifted preference towards sweet (Fig. 2d right panel), indicating that NPF neurons receive food-preference-relevant Lk and dopaminergic inputs. RNAi knockdown of DopEcR dopamine receptor in DH44 neurons shifted preference towards bittersweet (Fig. 2e right panel), indicating that DH44 neurons receive food-preference-relevant dopaminergic inputs.

Importantly, like optogenetic inhibition of FBl6 neurons (Fig. 3a left panel, Supplementary Fig. 3b left panel), RNAi knockdown of Lkr, AstA-R1, or DH44-R1 receptors in FBl6 neurons using either of two distinct GAL4 cell-specific driver lines also shifted preference towards bittersweet (Fig. 3a right panel, Supplementary Fig. 3b right panel). These findings indicate that FBl6 neurons are directly modulated by these three neuropeptides to affect food choice similar to optogenetic inhibition of FBl6 neurons. Furthermore, direction of change in

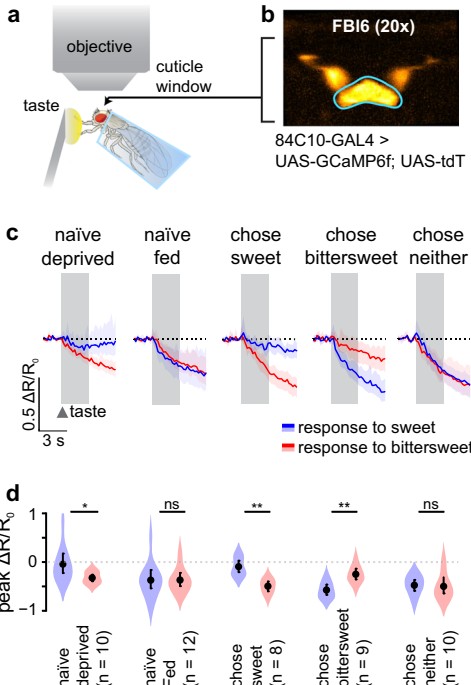

**Fig. 4 Neural activity in FBl6 encodes food choice. a** In vivo $Ca^{2+}$ imaging of taste responses in FBl6 neurons of flies in different hunger states making different sweet-bittersweet decisions. Sweet or bittersweet taste stimuli are applied to the forelegs, and changes in $Ca^{2+}$ responses are measured ratiometrically in FBl6 using GCaMP6f and tdTomato. **b** Neuronal expression in FBl6 reported by tdTomato. Region of interest for fluorescence measurement is outlined in cyan. 84C10-GAL4 driver line is used to specifically target strong fluorescent protein expression in FBl6 neurons[106,107] (Supplementary Fig. 3b–d). **c** Ratiometric calcium responses, $\Delta R/R_0$, of flies with different hunger state and decision outcomes. Sweet (50 mM sucrose) and bittersweet (500 mM sucrose + 1 mM quinine) taste stimuli from the equal-preference condition were applied for 3 s, and neural response was quantified for 4 s post-stimulus application. Taste stimulus application is indicated by gray background region. FBl6 neurons were strongly inhibited by the behaviorally rejected taste stimulus. Specifically, FBl6 neurons of flies that chose sweet were inhibited by bittersweet taste, while FBl6 neurons of flies that chose bittersweet were inhibited by sweet taste. **c** Calcium activity trace depicts mean $\Delta R/R_0 \pm$ 95% CI. **d** Peak $\Delta R/R_0$ shows significant difference between responses to rejected vs. chosen taste within each fly condition. $p < 0.05 = *$ (see Supplementary Table 1 for details on statistics). Points on graphs depict mean ± 95% CI, with violins depicting full data distribution.

food preference induced by receptor RNAi in FBl6 mirrors direction of change in food preference induced by optogenetic inhibition of the corresponding neuropeptide neurons. First, both AstA-R1 receptor RNAi in FBl6 neurons (Fig. 3a right panel, Supplementary Fig. 3b right panel) and optogenetic inhibition of AstA neurons (Fig. 2c left panel) shifted food preference towards bittersweet. Second, both DH44-R1 receptor RNAi in FBl6 (Fig. 3a right panel, Supplementary Fig. 3b right panel) and optogenetic inhibition of DH44 neurons (Fig. 2d right panel) shifted preference towards bittersweet. Finally, both Lkr receptor RNAi in FBl6 (Fig. 3a right panel, Supplementary Fig. 3b right panel) and optogenetic inhibition of Lk neurons (Fig. 2b right panel) shifted preference towards bittersweet. Since inhibition of AstA, DH44, and Lk neurons is equivalent to receptor RNAi of these neuropeptides in target neurons such as FBl6, and both of these manipulations produce the same shift in preference for bittersweet as optogenetic inhibition of FBl6 neurons alone, these

findings strongly suggest that inhibition of AstA, DH44, and Lk neurons results in inhibition of FBl6 neurons. The receptors for AstA, DH44, and Lk are GPCRs that potentially couple to distinct G proteins in FBl6 neurons, which in turn interact with different effector molecules to produce distinct downstream cellular responses. For example, AstA-R1 human homolog galanin receptor is coupled to $G_i$ and galanin binding initiates different intracellular signaling pathways in different target tissues[23]. DH44-R1 belongs to CRF-like secretin family of GPCRs that stimulate adenylate cyclase and can activate multiple $G_\alpha$ subunits[24], and *Drosophila* Lkr couples to $G_q$ signaling pathways[25,26]. These distinct G protein coupled downstream signaling pathways can maintain identities of signals encoded by the neuropeptides for integration in FBl6 neurons by coupling to ion channels, neurotransmitter receptors, and other biophysical and biochemical effectors to differentially modulate cellular excitability and synaptic transmission. RNAi knockdown of dopamine receptors in FBl6 had no effect (Fig. 3a right panel). While FBl6 neurons receive synaptic input from dopaminergic neurons[27,28] that regulate sleep[27,28] and ethanol preference[29], interestingly, direct dopaminergic input to FBl6 through dopamine receptors did not influence food choice (Fig. 3a right panel). Instead, we hypothesize that the dopaminergic neurons we identified regulating food choice modulate the activity of AstA, DH44, and NPF neurons (Fig. 2b–e), and thereby indirectly influence FBl6 neurons to modulate sweet-bittersweet choice.

This matrixed chemoconnectomics strategy mapped the neuromodulatory connections between nodes in the ensemble to control choice, and uncovered FBl6 as a previously unknown convergence node that is well positioned to integrate sensory, metabolic, and experiential information for decision making.

**Fan-shaped body neurons encode choice.** Value estimates of internal state variables, such as hunger, external environmental cues, appetitive or aversive value of food (i.e., valence), and past sensorimotor experience are integrated with one another and transformed into behavioral choice. This raises the key question whether FBl6 neurons compute value estimates or, instead, integrate these value estimates to encode choice. If FBl6 neurons encode or estimate value of metabolic parameters such as hunger or satiety, manipulating their activity would be expected to directly influence feeding behavior. To test this, we first quantified the proportion of flies that consumed food during optogenetic activation or inhibition of FBl6 neurons. During FBl6 neural activity manipulation, most food-deprived flies consumed food while most fed flies did not (Fig. 3b, Supplementary Fig. 3c, Supplementary Table 1), revealing that FBl6 activity manipulation neither prevented food consumption by food-deprived flies nor induced food consumption by fed flies. These results demonstrate that hunger state is not affected by FBl6 neural activity. Next, we quantified food consumption per fly during optogenetic activation or inhibition of FBl6 neurons and found no effect on total amount of food consumed per fly (Fig. 3c, Supplementary Table 1). These manipulations also did not affect the quantity of sweet versus bittersweet food consumed per fly within each condition (Fig. 3d, Supplementary Table 1), again demonstrating that FBl6 neural activity does not affect hunger state. Rather, the shift in food preference induced by optogenetic inhibition of FBl6 (Fig. 3a left panel, Supplementary Fig. 3b left panel) was due to an increase in the proportion of flies consuming bittersweet over sweet food, and not each fly consuming a larger quantity of bittersweet food. Taken together, these results demonstrate that FBl6 does not encode or affect metabolic signals of hunger or satiety.

If a neural population encodes reward or punishment, then manipulating the activity of that population could directly shift a

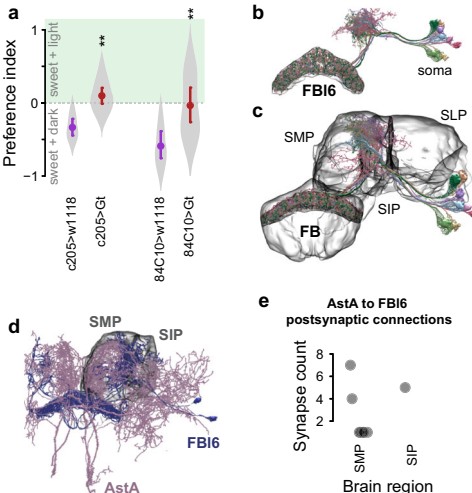

**Fig. 5 FBl6 neural activity is context-dependent. a** Food preference index of flies with spatially restricted persistent optogenetic inhibition. FBl6 neurons are optogenetically inhibited in one half of the arena that contains only sweet food on both sides. This inhibition abolishes aversion for bright green light by shifting food preference away from dark half of the arena. Points on graphs depict mean ± 95% CI, with violins depicting full data distribution. **b** EM reconstruction of FBl6 neurons shows neural projections restricted to layer 6 of the fan-shaped body. EM reconstruction of traced FBl6 neurons from the Janelia hemibrain dataset was performed using neuPrint and NeuronBridge for neurons targeted by 84C10-GAL4 and embedded in the surface representation mesh of standardized FBl6 brain region. **c** EM reconstruction of FBl6 neurons targeted by 84C10-GAL4 was performed as previously and embedded in the surface representation mesh of standardized whole fan-shaped body (FB). FBl6 neural projections to superior medial protocerebrum (SMP), superior intermediate protocerebrum (SIP), and superior lateral protocerebrum (SLP) higher brain regions are shown embedded in the surface representation of these standardized higher brain regions. **d** AstA neurons make postsynaptic connections with FBl6 neurons in SMP and SIP brain regions. Surface representations of brain regions where AstA neurons make postsynaptic connections with FBl6 neurons are shown in gray. EM analysis of traced AstA neurons that make postsynaptic connections with FBl6 neurons was performed in neuPrint. **e** Quantification of number of postsynaptic connections from AstA to FBl6 neurons. Each gray dot represents the number of synaptic connections between one AstA and one FBl6 neuron, separated by brain region on the x-axis where these connections are made.

decision without integrating value estimates or encoding choice. To test whether FBl6 activity is intrinsically rewarding or punishing, we illuminated half the decision arena with either red (for optogenetic activation) or green (for optogenetic inhibition) light in the absence of food and quantified preference for the illuminated versus dark sectors. Flies expressing optogenetic actuators CsChrimson or GtACR1 in FBl6 did not alter their preference for illuminated versus dark sectors compared to non-expressing controls (Fig. 3e, Supplementary Fig. 3d), demonstrating that FBl6 activation or inhibition is intrinsically neither rewarding nor punishing.

Animals accumulate information about past experience to inform future decisions. We hypothesized that FBl6 integrates internal hunger state and external food-related value estimates with experiential information to drive decisions. To understand how past experience affects FBl6 activity, we recorded FBl6 neural taste responses in flies that had different food-related experiences. Equal-preference taste stimuli were presented to forelegs of the fly (Fig. 4a) while measuring intracellular $Ca^{2+}$ signals ratiometrically using GCaMP6f[30] and tdTomato (Fig. 4b) expressed in FBl6

(Fig. 4b). First, we tested the effect of hunger on FBl6 taste responses in naïve flies, i.e., flies that had not experienced the decision task at all. FBl6 neurons of naïve food-deprived flies were strongly inhibited by the bittersweet stimulus, but not sweet (Fig. 4c, d, "naïve deprived"). Flies innately find bittersweet food aversive[31], and reduce proboscis extension response to and consumption of bittersweet mixtures[32–34]. Thus inhibitory taste responses in FBl6 to the first encounter with bittersweet food may represent rejection of this option. In contrast to hungry food-deprived flies, fed flies reject both foods in the decision task (Fig. 3b, Supplementary Fig. 3c). If FBl6 neural activity represents rejected choice, then we expect FBl6 inhibition by both sweet and bittersweet stimuli in naïve fed flies. Indeed, FBl6 neurons of naïve fed flies were strongly inhibited by both bittersweet and sweet stimuli (Fig. 4c, d, "naïve fed"). Basal FBl6 neural $Ca^{2+}$ levels in the absence of taste stimulation was indistinguishable between naïve hungry and fed flies (Supplementary Fig. 4e–h). Importantly, this is the first observation of gustatory responses in the *Drosophila* fan-shaped body, or any other part of the central complex.

If, as in naïve flies, FBl6 neural activity also represents rejected choice in food-deprived flies that have had the opportunity to decide between sweet and bittersweet in the equal-preference condition (50 mM sucrose vs. 500 mM sucrose + 1 mM quinine), then we expect this experience and the decision outcomes to affect FBl6 taste responses. Specifically, we predict FBl6 neurons of each fly to be strongly inhibited by the rejected choice for that fly, based on its individual decision. To test this, we measured FBl6 neural response to sweet and bittersweet tastes in flies that had experienced the equal-preference decision assay. Almost all flies visit both the sweet and bittersweet food sectors (Supplementary Fig. 1h) during the decision task (Supplementary Fig. 1g–i), transitioning from one sector to the other multiple times over the 5 min choice assay. These transits provide flies with sensory information about both food options via gustatory receptors on their legs, since flies are confined to stand on the food surface. The food option that an individual fly consumes is recorded by scoring the color of its abdomen at the end of the assay. Almost all individual flies consume either sweet or bittersweet food only, meaning that under the equal-preference condition approximately 50% of flies ate sweet and the other 50% ate bittersweet food resulting in an equilibrium for the population of flies within a trial. From this equal-preference trial, for imaging we selected individual flies that consumed sweet only, bittersweet only, or no food. Since these individual flies chose only one type of food, we were able to map each fly's subsequent FBl6 neural responses to its chosen and rejected food options. Consistent with our hypothesis, when taste responses were subsequently measured in food-deprived flies that chose sweet food in the equal-preference decision task, FBl6 neurons were strongly inhibited by the rejected bittersweet stimulus, but not by the chosen sweet (Fig. 4c, d, "chose sweet"). Furthermore, FBl6 neurons of flies that chose bittersweet food were strongly inhibited by the rejected sweet stimulus, but not by the chosen bittersweet (Fig. 4c, d, "chose bittersweet"). Finally, FBl6 neurons of flies that chose neither option, i.e., rejected both, were strongly inhibited by both bittersweet and sweet stimuli (Fig. 4c, d, "chose neither").

To further test whether FBl6 neural response to the previously tested sweet and bittersweet taste sensory cues is absolute or context-dependent, we presented food-deprived flies with a no-choice assay in which only one type of food (either sweet or bittersweet) was present. We then measured FBl6 neural responses to both sweet and bittersweet taste stimuli in the flies that consumed the only available food option in this no-choice context (Supplementary Fig. 4a–d). FBl6 neurons were inhibited by neither the taste of food that flies experienced and consumed

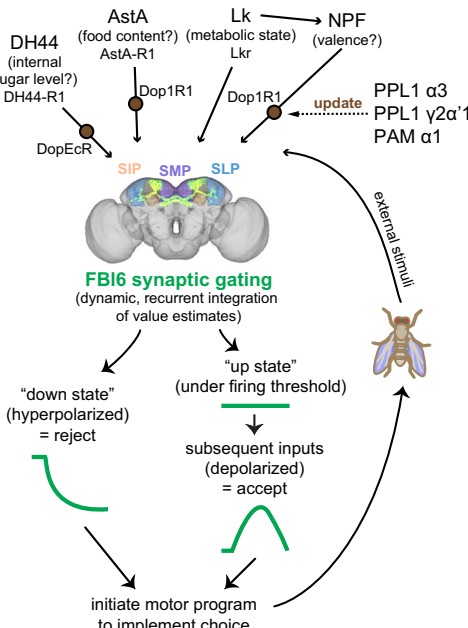

**Fig. 6 Proposed synaptic gating model for recurrent integration of updated input signals by FBl6 neurons for decision making.** Upstream neuropeptidergic neurons assess value estimates of metabolic state (Lk), internal sugar level (DH44), available food content (AstA), and food valence (NPF), and project to SMP, SIP, and SLP higher brain regions. In these brain regions, dopaminergic neurons recurrently modulate neuropeptidergic neurons to update value estimates encoded by them when animal navigates in its environment, senses, and consumes food. Context-dependent updated value estimates are dynamically received from neuropeptidergic neurons by FBl6 neurons in the SMP, SIP, SLP regions. FBl6 neurons are bistable, transitioning between a hyperpolarized "down state" and a resting state associated "up state". When FBl6 neurons are in "down state," choice outcome is rejection. When FBl6 neurons receive inputs to reach "up state" with membrane potential just under firing threshold, subsequent inputs are integrated non-linearly. If this non-linear summation crosses the threshold to depolarize FBl6 neurons, choice outcome is acceptance. In sum, FBl6 neurons dynamically integrate and gate the flow of information to downstream circuits, initiating motor programs that implement the choice outcome. This transient dynamic representation of choice is constantly updated over time with animal's experience, internal state, and sensory environment. Dynamic updating of these signals is achieved by extensive recurrent circuitry between the inputs and outputs of the decision ensemble. Standardized brain with SMP, SLP, SIP brain regions and neurons expressed by 84C10-GAL4 were created at VirtualFlyBrain.org.

during the no-choice task, nor by the new taste stimulus that flies did not experience before (Supplementary Fig. 4a–d). Since these flies experienced food in a no-choice context, they had a different prior experience from the flies that experienced the conflicting two-choice task, which modified their FBl6 neural response to these stimuli. Flies that consumed the only food option available to them showed no response to the consumed taste stimulus. Thus, it is possible that lack of response to a novel food taste reflects acceptance of that novel food option. In summary, strong inhibitory FBl6 neural responses to taste stimuli are context-dependent and correspond to rejected food choice, suggesting that context-dependent temporally coupled inhibitory FBl6 responses are the neural representation of rejected behavioral food choice.

FBl6 neurons are strongly inhibited by a single stimulation with rejected food choice (Fig. 4c, d), this inhibition is context-dependent, and persistent sensory-stimulus independent optogenetic inhibition of FBl6 neurons shifts food preference to bittersweet (Fig. 3a left panel, Supplementary Fig. 3b left panel). We thus hypothesized that imposing unnatural, temporally and spatially homogenous optogenetic inhibition of FBl6 throughout the decision task disrupts value integration and decision making. This imposed activity pattern, that does not take into account the sensory environment and current value estimates of the fly in any given moment, results in the hungry fly defaulting to higher calorie bittersweet food. To test this further, we optogenetically inhibited FBl6 neurons on only one side of the arena while presenting flies with the same sweet food on both sides. We found that control flies not expressing GtACR1 avoided the lit side of the arena indicating intrinsic avoidance of bright green light[35]. In contrast, persistent FBl6 optogenetic inhibition abolished this bright green light avoidance, resulting in equal preference for food on both sides (Fig. 5a, Supplementary Table 1). While spatially and temporally homogeneous optogenetic inhibition of FBl6 neurons shifted preference to bittersweet (Fig. 3a left panel, Supplementary Fig. 3b left panel), spatially restricted optogenetic inhibition shifted preference to sweet (Fig. 5a). Congruent with context-dependent FBl6 neural response to taste stimuli (Fig. 4c, d, Supplementary Fig. 4a–d), optogenetically inhibiting the same FBl6 neurons in different behavioral contexts produced different choice outcomes.

These findings are inconsistent with predictions of a simplistic ON/OFF sensorimotor model in which FBl6 neurons would turn feeding on and off on command independent of context. Instead these findings support a model in which the activity of FBl6 neurons is transient, dynamic, and highly state- and context-dependent. Similar to mammalian cortical gatekeeper neurons[36–38], FBl6 neurons could act as a gatekeeper that integrates converging inputs from upstream neuropeptidergic neurons using non-linear summation (Fig. 6). If the activity of FBl6 neurons were forced into an unnatural "down state" using optogenetic inhibition, the integration of inputs to FBl6 neurons would be disrupted. Under such a condition, hardwired innate inputs could be shunted to bypass the gatekeeper circuit motif resulting in unexpected behavioral outcomes that would not take into account adaptive flexible input variables such as the context in which flies choose food options.

Based on the effects of receptor RNAi manipulations in FBl6 neurons (Fig. 3a, Supplementary Fig. 3b), we hypothesized that FBl6 neurons receive as inputs food and internal state related parameters encoded by the upstream neuromodulatory neurons; directly from AstA, DH44, and Lk neurons, and indirectly from NPF and dopamine neurons. To assess whether AstA, DH44, and Lk neurons form direct synaptic connections with FBl6 neurons (Fig. 5b, c), we queried the publicly available *Drosophila* hemibrain EM connectome dataset[39] for these neurons. We identified two traced AstA neurons both of which form postsynaptic connections with FBl6 neurons in the SMP and SIP higher brain regions (Fig. 5d, e), with one of the AstA neurons forming up to seven connections with one of six different FBl6 neurons. Since Lk and DH44 neuron tracing data is currently not available, we could not determine if Lk and DH44 neurons also form synaptic connections with FBl6 neurons. The signals that FBl6 receives from these upstream modulatory neurons are then integrated by FBl6 in a state- and context-dependent manner to generate a transient and dynamic representation of choice, which is relayed to downstream motor circuits (Fig. 6).

## Discussion

Animals make decisions about which foods to consume by integrating internal physiological state signals with external sensory

cues. Here we delineated a neuronal ensemble in *Drosophila* that underlies food-related decision making during sensory conflict between sweet and bittersweet food choices (Fig. 1). Activating or silencing particular nodes in this ensemble shifts the decision balance between sweet and bittersweet choices (Figs. 2, 3). Inputs encoding hunger state, taste identity, and past experience converge on FBl6 neurons (Fig. 4), which integrate these inputs to generate a neural representation of food choice that drives motor outputs for foraging decisions (Fig. 5).

Animals assess and assign value estimates to internal state and external environmental parameters before integrating these estimates to guide adaptive decision making. While the mechanistic basis of integration of internal state and external stimuli on a molecular or biophysical level in the *Drosophila* higher-order brain remains unclear, prior studies reveal some cell-populations and brain regions that are involved in various decision making contexts. Various neuromodulatory and other neurons regulate hunger dependent food intake[40–46], hunger dependent odor encoding and food search[47–49], reward[13,50–52] or punishment[53], memory[12,13,52], and internal state for courtship[54–56]. The mushroom body is an insect central brain region involved in gustatory learning and memory[31,57] and valence encoding[58], and is thought to be a major center controlling higher-order behaviors[47], including associative learning[59–61]. The insect central complex is an evolutionarily conserved central brain region whose ellipsoid body and protocerebral bridge sub-regions have been implicated in navigation[62–69] and sleep[70–72]. The central complex fan-shaped body, a sub-region tiled by columnar and tangential neurons[73], has been implicated in sleep[18,27,28,74] and ethanol preference[29,75]. Various neuromodulators[19], their receptors[16–18], and dopaminergic inputs[20,21] co-localize in the layers of fan-shaped body.

Upstream neuromodulatory subsets in the decision ensemble we identified have known roles in hunger dependent food intake[40,43,45,76], reward[77], valence[10,12], and long-term memory[10,12,13]. Since under natural foraging conditions, flies conduct food search while walking and flying, it is expected that sensory information about food seeking during flight would also be integrated in this ensemble. Consistently, we also identified flight-promoting dopaminergic neurons involved in food seeking behaviors[11,61]. Hungry flies take longer to identify a food source when specific dopaminergic neurons (PPL1 γ2α′1) are inhibited[11,61]. Since FBl6 neurons are involved in accepting and rejecting food options, integrate hunger state to inform this decision, and affect food-related behavior in a no-choice context (Fig. 5a, Supplementary Fig. 5), it is conceivable that these neurons could be involved in making a decision to not search for food in the absence of a choice when flies are sated.

These modulatory neurons are well positioned to estimate value of features in the sensory environment and internal state that can be updated in a state- and context-dependent manner. For example, AstA neuron activity influences relative carbohydrate and protein preference[40], while DH44 neurons sense sugars[76] and amino acids[45]. Thus, AstA and DH44 neurons could convey food identity information to FBl6. NPF neuron activation is inherently rewarding[77], and thus could convey food valence information. Lk neurons have been implicated in nutrient sensing[43,78], and their activity regulates feeding in food deprived flies (Fig. 2a, b), suggesting that internal metabolic state information could reach FBl6 through Lk/Lkr signaling. Neuropeptides perform diverse functions that are highly state- and context-dependent. For example, Lk and Lkr mutants have been shown to have opposite effects on proboscis extension response in response to sugar in hungry flies[43], which have been hypothetically attributed to different receptor or downstream signaling pathways. Mammalian NPY producing AgRP neurons are also known

to have highly state- and context-dependent roles in feeding behaviors. Recent studies have challenged the textbook model and shown that AgRP neurons are inhibited within seconds by sensory detection of food even though on the longer timescale AgRP neurons are activated by energy deficit and promote food consumption[79–81]. Similarly, mammalian hypothalamic POMC neurons were initially thought to encode satiety, but recent studies have found that their activation can also promote feeding[82]. The effects that different neuropeptidergic neurons have in the food-related choice assay employed here are thus concluded to be specific to the context of sweet-bittersweet gustatory decision making.

FBl6 neurons project their axons into the fan-shaped body[20,21], have dense dendritic projections in the SMP, and have sparse dendritic projections in SIP and SLP[20,21] (Fig. 5c). AstA neurons form postsynaptic connections with FBl6 neurons in SMP and SIP regions (Fig. 5d, e), and Lk[78,83] and DH44[44,84] neurons have extensive projections in the SMP, SLP, and SIP regions. It is probable that FBl6 neurons receive decision relevant signals from these upstream neuropeptidergic neurons in these higher brain regions (Fig. 6). Dopaminergic subsets involved in aversive memory and modulation of flight and feeding (PPL1 γ2α′1)[10,11], taste conditioning (PPL1 α3)[12], and long-term memory (PAM α1)[13] also affect food choice in the decision assay (Fig. 2a), and could provide signals for predicting and updating value estimates in working memory, analogously to primate dopaminergic ventral tegmental area[85]. Since the identified dopaminergic neurons project to SMP, SLP, and SIP regions, where the neuropeptidergic neurons also arborize and connect to FBl6 neurons, it is conceivable that it is in these same higher brain regions that the identified dopaminergic neurons directly modulate neuropeptidergic neurons (Fig. 2b–e) to update value estimates encoded by them.

As the value estimates are updated in a state- and context-dependent manner, they are dynamically received by FBl6 neurons. We hypothesize that, similar to mammalian nucleus accumbens and other cortical circuits involved in gating the flow of information to cortex for attention and decision-making[36–38], FBl6 neurons are bistable and oscillate between two states that are characterized by different values of membrane potential. According to the synaptic gating model[36–38], in the "down state", FBl6 neurons would be hyperpolarized (i.e., closed gate) leading to rejection of an option. To reach the "up state", i.e., a membrane potential just below the neuron's firing threshold, FBl6 neurons would have to receive input from another neuron. Only when FBl6 neurons are in this "up state" (i.e., open gate) can subsequent inputs trigger action potentials leading to acceptance of an option influenced by these inputs FBl6 neurons could act as a gatekeeper circuit motif that non-linearly integrates converging inputs and gates the flow of information to downstream circuits to initiate motor programs implementing the decision (Fig. 6). This transient dynamic representation of choice is updated as the value estimates change over time with animal's experience, internal state, and sensory environment. Such transient dynamic neural activity representing a constantly updated choice would require extensive recurrent circuitry between the inputs and outputs of the decision ensemble. Congruently, SMP, SLP, SIP, and FB brain regions have extensive recurrent circuits[73] that may be involved in such dynamic updating and integration for decision making.

Other neuromodulatory neurons, second-order sensory projection neurons, and interneurons also likely interact with this decision ensemble to convey gustatory and other sensory inputs. There are no known projections of primary gustatory sensory neurons or second-order taste neurons[86–89] to the fan-shaped body. Food preference was affected neither by acute optogenetic

nor by chronic inhibition (using tetanus toxin light chain to impair vesicle docking) of the only known taste projection neurons (TPN3) that relay bitter taste information to SLP and are essential for conditioned taste aversion[88] (Supplementary Fig. 2c). However, SMP, SLP, and SIP higher brain regions, to which FBl6 neurons also densely project (Fig. 4g), have been implicated as target areas for second-order taste neurons[90]. It thus is possible that sweet and bitter taste information reaches the fan-shaped body through second-order taste projection neurons that terminate in these higher brain regions. EM connectome data show that outputs from different areas of the subesophageal zone (SEZ), the brain region to which most gustatory receptor neurons converge, tile the SMP and SIP where they form synaptic connections with different dopaminergic neurons, including PPL1-α3 that we have identified as influencing sweet-bittersweet decision making (Fig. 2a)[91]. It is possible that taste identity information is conveyed to FBl6 by these dopaminergic neurons that receive inputs from SEZ output neurons, respond to sweet[52] and bitter[12] tastes, are involved in taste conditioning[12,52], and shift preference in our decision assay (Fig. 2a, PAM-α1, PPL1-α3).

Mammalian studies provide converging evidence for multiple interconnected networks in frontal cortex and basal ganglia that compute and store value estimates of sensory environment and motor events in that environment required for decision making[85,92]. The neural ensemble described in this study is an analogous framework of interconnected networks for potentially storing, computing, and updating value estimates that are likely integrated by FBl6 neurons. While decision making theories in mammals have focused on how values are represented in the brain[85,92], mechanisms by which the brain integrates value information to make decisions remain unclear[93,94]. Changes in neural spiking pattern are thought to underlie sensory information accumulation and integration, with a decision being made when neural firing rate reaches a threshold[95–97]. Persistent neural activity and synaptic plasticity changes are thought to underlie storage of sensory information and past choice history to guide adaptive decisions[96]. Future work is required to test the proposed hypotheses of specific roles for each node in the fly decision ensemble, how upstream inputs are integrated in FBl6, how this integration is transformed into the representation of choice, and how downstream motor circuits implement the foraging decision.

## Methods

**Fly husbandry.** Flies were cultured on standard cornmeal medium on 12:12 light:dark cycle at 25 °C. *w1118* lab stock was used as wild type. All other genotypes and their sources are described in Supplementary Table 2. 2–5 day old flies were wet starved for 21 h on wet Kimwipe with 1.5 ml distilled water. Experiments in Fig. 1 and Supplementary Fig. 1 we conducted after the food deprivation duration mentioned in those figures. RNAi experiments using 84C10-GAL4 were performed after 29 h of food deprivation. For optogenetic experiments, flies were food deprived for 21 h before testing on 0.4 mM all-*trans* Retinal (Cayman Chemicals) in 1% agar. Flies for RNAi knockdown and their controls were moved to 28 °C for 21 h the day before testing, i.e., during food deprivation, to induce strong RNAi. RNAi control was created for each GAL4 line by crossing the respective GAL4 to UAS-Valium (see Supplementary Table 2). All RNAi lines that we used were from Harvard TRiP project[98,99] and have been validated by independent groups (see Supplementary Table 2). Flies for simultaneous optogenetic and RNAi experiments were created using the genotypes mentioned in Supplementary Table 2. All experiments were conducted at Zeitgeber Time 3–6.

**Two-choice assay and optogenetics.** Sweet foods were made with different concentrations of sucrose and bittersweet foods with 500 mM sucrose (Sigma) and 1 mM quinine (Alfa Aesar or Beantown Chemicals, CAS#207671-44-1) dissolved in 1% agarose (AmericanBio) made in distilled water. 0.04% w/v red dye (Sulforhodamine B, MP Biochemicals, CAS# 3520-42-1) and 0.02% w/v blue dye (Erioglaucine A, Alfa Aesar, CAS# 3844-45-9) were used for food coloring. Dye colors were alternated between sweet and bittersweet foods for each condition and there was no preference for one dye over the other at the concentrations used. Fly arenas were prepared by pouring agarose based foods in two-compartment petri-dishes (90–100 mm diameter) from Kord Valmark, EMS, or Fisher Scientific. Because of a

thin physical barrier between the two compartments in the arena there was no diffusion between the two foods. Groups of 20–35 flies were aspirated and introduced into the arena 5–10 s before the start of the experiment. All experiments were conducted in dark so that there was no effect of food color on preference. Arenas were placed on a platform with IR backlight for video recording using a Flea Pointgrey camera (FL3-U3-20E4C/M) and Pointgrey software was used to acquire videos at 15 fps. For optogenetics, we used high-power LEDs (Luxeon) placed adjacent to backlight IR LEDs (based on Janelia ID&F design) of 627 nm (for CsChrimson) and 520 nm (for GtACR1) that were controlled using Arduino Uno. For optogenetic screen, both red and green lights were pulsed at 100% max intensity, 50 Hz, 25% duty cycle. For follow up experiments, CsChrimson experiments were conducted at 25% max intensity; GtACR1 follow up was done at screen condition. Light was pulsed for the entire duration of the experiment. At the end of the experiment, flies were anaesthetized using $CO_2$ and their abdomen color was recorded under a dissection microscope. Preference index (PI) was calculated as (no. of sweet food flies + 0.5 no. of both food flies) − (no. of bittersweet food flies + 0.5 no. of both food flies)/no. of total flies that ate, where negative PI would mean that more number of flies ate bittersweet food. We also recorded the number of flies that did not consume either food in the assay by counting the number of flies that had neither food dye in their abdomen.

**Food intake quantification.** Food intake was quantified using spectrophotometry[100,101]. Specifically, after recording belly color flies were frozen in 1.5 ml Eppendorf tubes at −20 °C until intake quantification (1–2 days). Flies from each trial were separately homogenized in distilled water (5 μl/fly) using a motorized pestle (BT Labsystems, BT703) for 1.5 min and centrifuged at $18,928 \times g$ for 5 min. Absorbance of the debris-cleared 2 μl supernatant was measured on NanoDrop 2000 Spectrophotometer (Thermo Fisher Scientific) at 565 nm (for red dye) and 630 nm (for blue dye). Flies that ate uncolored 1% agarose with 50 mM sucrose were used as blank for baseline control. Red and blue dye concentrations were interpolated using their respective standard curves (GraphPad Prism) acquired from serial dilutions of single dyes in distilled water. Since we knew the number of flies that ate each color per trial, we could calculate per fly blue and red concentrations in the same solution.

**Calcium imaging and data analysis.** 3–5 day old flies (naïve or after two-choice assay) were aspirated and positioned in a custom made fly holder in which they were glued using two-part transparent epoxy (Devcon). Only the top of fly head (for imaging) and the forelegs (for taste delivery) were outside the holder, while the rest of the fly, including proboscis were restrained in the fly holder. No anesthesia was used. A small piece of head cuticle was dissected and air sacs removed using a 30-gauge syringe needle and fine forceps, immediately followed by sealing the head capsule with a translucent surgical silicone adhesive (Kwil-Sil, WPI). Dissected fly was then placed in a humidified chamber for 15 min recovery before imaging. Flies that experienced the two-choice assay were tested within 20 min to 1.5 h after the assay. Fed flies ate regular cornmeal food, were never food deprived, and didn't experience the two-choice assay. Flies that experienced the two-choice assay were food-deprived for 21 h and chose to eat agar based sweet or bittersweet food (not regular cornmeal food) for only 5 min during the assay.

Calcium imaging was performed on a Zeiss Axio Examiner upright microscope with 20× air objective and a Colibri module for LED control. tdTomato was excited at 555 nm (80% intensity) and GCaMP6 at 470 nm (100% intensity). An image splitter (Photometrics DV-2) was used to split red and green channels and acquire simultaneous images for tdTomato and GCaMP6 using a Hamamatsu ORCA-R2 C10600 camera. Images were acquired using Zen software at 5 fps with variable baseline (3–10 s required for stable tastant delivery) followed by tastant application to the forelegs, using a syringe, for 3 s and 4 s of no tastant. Excess tastant was wicked from the forelegs using absorbent tissue paper between each application. 10 s inter-trial interval was used during which all lights were off. Water was always applied first, followed by either sweet or bittersweet tastant. Sequence of sweet and bittersweet was alternated between flies. Sweet: 50 mM sucrose in distilled water; bittersweet: 500 mM sucrose + 1 mM quinine in distilled water. For flies that chose sweet in the two-choice behavior assay only trials with sweet as the first tastant were averaged and for flies that chose bittersweet, only trials with bittersweet as the first tastant were averaged.

Pixel intensities were extracted in Fiji followed by data analysis in MATLAB, both using custom written code. After background subtraction using Fiji's rolling-ball method (20 px), ROIs were manually drawn and saved on the tdTomato image, and superimposed on the GCaMP image (both reporters were expressed in the same neurons using the same driver) for mean ROI pixel intensity extraction. The saved intensity signals were then analyzed in MATLAB. tdTomato and GCaMP traces were individually corrected for photobleaching by fitting a single exponential function. Corrected GCaMP trace was then divided by the corrected tdTomato trace to obtain the ratiometric fluorescence trace ($R$). For relative fluorescence fold change ($\Delta R/R_0$) determination, baseline fluorescence ($R_0$) was calculated by averaging R over 2 s preceding tastant application. Peak $\Delta R/R_0$ was calculated during 4 s following tastant application.

**Animal tracking**. Flies were tracked using Caltech FlyTracker[102]. Tracked data were analyzed and visualized using custom MATLAB code.

**EM reconstruction and analysis**. Electron microscopy images were reconstructed from publically available Janelia FlyEM hemibrain data using neuPRINT explorer[39,103]. Neuron identities were confirmed in NeuronBridge[104] by cross-referencing EM traced FBl6 neurons matched with light microscopy images of 84C10-GAL4 from FlyLight[105]. FBl6 mesh, whole FB mesh, and SMP, SIP, and SLP brain region meshes were used to depict brain regions with neural projection areas. Synaptic connections were analyzed using neuprint-python.

**Statistics**. All data were plotted in either Python or MATLAB using custom written code. Statistics were carried out either in GraphPad Prism, Python, or MATLAB. If all data passed Kolmogorov-Smirnov normality test, ANOVA was conducted, otherwise Kruskal–Wallis test was conducted, both followed by appropriate post-hoc tests. Details of statistics for each figure are provided in Supplementary Table 1. Sample sizes are reported in parentheses next to dataset name in Supplementary Table 1.

**Reporting summary**. Further information on research design is available in the Nature Research Reporting Summary linked to this article.

## Data availability

Further information and requests for resources and reagents should be directed to and will be fulfilled by the Lead Contact, Dr. Michael N. Nitabach (http://michael.nitabach@yale.edu). All data generated in this study are provided in the Source Data file. Source data are provided with this paper.

## Code availability

Custom code that support the findings from this study are available from the Lead Contact upon request.

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

## Acknowledgements

We thank Peter Niesman for help with calcium imaging data collection and Jason Braco for helpful discussions. Yichen Luo from John Carlson's lab provided useful information for design of the taste delivery imaging rig. Tanya Wolff provided insights into fan-shaped body neuroanatomy. We also thank Gerry Rubin and Tanya Wolff for sharing unpublished fly lines ss00208 and ss00225. These studies were supported in part by the National Institute of General Medical Sciences, NIH (R01GM098932).

## Author contributions

P.F.S. conceptualized the study, designed and performed experiments, and analyzed data; L.Y.M. performed experiments; P.F.S. and M.N.N. interpreted data; P.F.S. and M.N.N. prepared the manuscript with input from all authors.

## Competing interests

The authors declare no competing interests.
