## [Peer Review File · Nature Communications]

REVIEWER COMMENTS

Reviewer #1 (Remarks to the Author):

In this manuscript, Sareen, McCurdy, and Nitabach seek to identify neurons and neuromodulatory systems involved in the integration of external food cues and internal states to perform adaptive feeding decisions. They first develop a two-choice feeding assay in which flies can choose to eat from two sides of a plate coated with different tastants (bittersweet vs various concentrations of sweet). They identified an ensemble of cell types – notably, Leucikinin neurons, Allatostatin neurons, NPF neurons, DH44 neurons, and aversive PPL dopamine neurons that influence feeding decisions and feeding amount. They also identify a set of layer 6 fan-shaped body neurons whose silencing biases the flies towards eating from the bittersweet area. The authors perform a very nice set of RNA knock-down experiments in these cells to show that the effects of manipulating peptidergic and neuromodulatory neurons are consistent with the effects of manipulating receptors for these signalling molecules in this downstream neuron. Finally, the authors use calcium imaging to show that the activity of L6 FB cells changes with feeding state, and that responses to the two tastants used in their study vary with which food the animal ate in the feeding assay. On the basis of these experiments, the authors claim that L6 FB neurons encode behavioral choice and integrate taste quality, choice outcome, and hunger state to influence behavioral decisions. These results are potentially novel and interesting, however there is an important confound that the authors need to address to support their claims.

Major issues:

1) Imaging experiments do not distinguish between choice encoding and sensory adaptation. A key claim of this paper is that the activity of FBL6 neurons reflects feeding decision of the fly. This conclusion was made because FBL6 neurons are inhibited when flies experience a food type that they previously rejected (sweet or bittersweet), or that they innately reject (bittersweet in naïve flies). An alternative explanation for this observation is that FBL6 neurons, or the circuitry upstream of FBL6, undergoes adaptation as a result of prolonged food exposure. In the choice paradigm used in this study, the flies likely experience/sense the ‘chosen’ food more than the ‘non-chosen’ food. If true, then unequal food exposure could cause greater adaptation towards the chosen food over the non-chosen food, and as a result, lead to different responses in the FBL6 neurons. To rule out adaptation as the reason for differential encoding of the “chosen” food, the authors should consider experiments putting flies on a purely sweet circular arena or a purely bittersweet circular arena for the same amount of time as in the choice assay, and then measuring FBL6 neural activity through Ca imaging. If responses are similar as in the choice assay, it would suggest that the differential encoding reflects simple sensory adaptation. Alternatively, if the differential encoding is not observed here, it would provide stronger evidence that the activity in FBL6 reflects a “decision” or “choice”.

2). A conceptual model linking the observed FBL6 activity to behavioral decisions is not presented.

The authors identified an ensemble of neurons that converge onto FBL6 cells to influence feeding decisions; however, they do not provide hypotheses about how these neuromodulators might generate the inhibition (or lack of inhibition) observed in FBL6 neurons under different conditions. It’s also unclear how inhibition or lack of inhibition in FBL6 neurons might influence feeding behaviors. A paragraph in the discussion, and perhaps a figure suggesting (1) how the various neuromodulators and neuropeptides might affect the physiology of FBL6, and (2) how FBL6 activity might influence behavioral decision through the fan-shaped body would help clarify the authors’ hypotheses and suggestions for future experiments.

3) The treatment of hunger and encoding of “internal state” is confusing. It is true that 84C10 neurons show different responses when starved or fed, but the main behavioral assay used in the paper does not shift its equilibrium point with hunger state and c205 manipulations do not alter feeding based on hunger state. The authors should clarify their language describing the circuits they found as encoding or integrating “internal state” to inform a decision. See several points below.

Additional comments:

The majority of the behavior was performed in a different Gal4 line than the imaging. I did not see a reference to Extended data Fig. 3 in the text (?) and so completely missed this important control on my first read. This needs to be prominently cited and clarified.

The section describing FBL6 activity in response to water was confusing and under-developed. Several claims are made about these responses which rest on speculation. The authors might either investigate their thirst hypothesis further or leave this section out. Although it is not necessary for the current manuscript, imaging from some of the upstream neuromodulatory neurons identified in this manuscript would clarify where the signals observed in L6FB emerge and what types of signal transformations occur between these populations.

Line 76: “This equal-preference point was identical at all of the tested food deprivation durations, suggesting an external taste sensation and internal hunger state equilibrium at this concentration ratio.”: Can you claim that the external taste sensations are in balance with internal hunger state if manipulating the hunger state does not alter the equal-preference point? The data suggests that the equal-preference point is invariant to internal hunger state and only depends on a balance of external sensory cues. See point (3) above.

Line 100: “While neural substrates integrating internal state and external stimuli in the *Drosophila* brain remain unclear, there are hints in the literature of cell populations and brain regions that could be involved.” I don’t think this is a fair statement. Numerous papers have explicitly examined this question in different behavioral contexts. For example, Sayin et al. 2018 explicitly identifies several MBONs, DANs, and octopaminergic neurons as encoding and integrating odor and taste stimuli to dynamically shape the intensity of odor tracking behavior in a hunger-dependant manner. Work from the Wang lab (Root and Wang 2011, Ko and Wang 2015) have identified molecular mechanisms at the olfactory periphery that allow hunger state to shape odor encoding. Vogt and Samuel 2020 show that in *Drosophila* larvae, hunger modulates a serotonergic neuron that alters the routing of odor information through the antennal lobe to influence behavioral decisions. In the courtship system, various neurons integrate internal state information to shape the intensity of courtship behaviors (Deutsch and Murthy 2020, Hindmarsh and Ruta 2020). These studies should be cited here.

Line 137: “Feeding suppression induced by Lk neuron activation implies a decrease in perceived hunger of food-deprived flies, consistent with the shift towards higher calorie bittersweet food by Lk neuron inhibition reflecting increased perceived hunger.” This logic does not seem clear. Hunger itself does not shift the animals’ preference for sweet vs bittersweet. See point (3) above.

Line 208: “Taken together these results demonstrate the FBI6 does not encode or affect metabolic signals of hunger or satiety” seems in conflict with line 218: “We hypothesized that FBI6 integrates internal hunger state and external food-related value estimates with experiential information to drive decisions”

Line 198: “FBI6 activity manipulation neither prevented food consumption by food-deprived flies nor induced food consumption by fed flies”. This finding is inconsistent with the claim made elsewhere that FBI6 integrates internal and external cues to generate a behavioral decision. (Related to point 3 above)

Line 225: “FBI6 neurons of naïve food-deprived flies were strongly inhibited by bittersweet

but not sweet.” It looks like there is some response to the sweet stimulus in this plot.
Line 227 “Flies often find bittersweet food aversive and thus inhibitory taste responses in FBL6 to bittersweet food may represent rejection of this option”. I don’t think that can be concluded in this experiment because the flies have not been exposed to any choice and there are many situations where these two stimuli produce a 50-50 choice in their assay.
Line 236: taste responses were subsequently measured in flies that chose sweet food in the decision task. How was choice quantified here? What were the criteria for having chosen sweet or bittersweet? What was the distribution of choices?
Line 286-288: DH44 and AstA neurons: how direct are the connections from these to FBL6?
Line 306: food preference was not affected by manipulations of TPN3: is there any olfactory component to the behavior?
Line 319: “Lack of FBL6 inhibition in response to a taste stimulus could represent acceptance of that option”. If FBL6 neurons are normally inhibited by tastants that they do not choose, how does inhibition of FBL6 with GPCR alter food preference towards bittersweet? (connected to comment 2 above—what is the conceptual model?)
Line 597: “3-5 day old flies (naïve or after two-choice assay)”: How much time passed between the two-choice assay and calcium imaging? What is the difference between “fed flies” and “two-choice assay” flies considering both groups recently ate food?
Fig2, panel b: how was lack of feeding quantified?
Fig 4, panel c: how does this compare to c205-Gal4?
Fig. 4, panel d: when do these responses return to baseline?
Fig. 4, panel d-e: why are the bittersweet responses in ‘chose bittersweet’ so variable compared to all other groups? Should increase the sample size slightly.

Reviewer #2 (Remarks to the Author):

The manuscript by Sareen et al., describes a set of very interesting and insightful experiments that help identify a novel region of the brain in decision making. It is well written for the most part. The authors begin with devising a robust choice-test that measures decision making in hungry flies. With this test they screened a series of neuronal classes that are likely to be implicated in decision making in the context of the taste stimuli and the internal state of the fly. They identify a limited and interesting set of neuropeptides (NPs; many of which have been implicated in feeding behaviour earlier), three sets of dopaminergic neurons and very interestingly a higher brain centre – the fan shaped body (FB) from the central complex. Using optogenetics and a knockdown approach for neuropeptide and dopaminergic receptors they identify putative pathways for information flow amongst the various neuronal classes. Similarly they place some NPs as directly regulating activity of FB neurons whereas others as sending information indirectly to the FB neurons. Finally, by directly measuring activity in the FB neurons during a choice test they show that increased activity correlates with the behavioural choice to a sensory stimulus. While the data are convincing, given the complexity of the subject there are several aspects that need addressing and clarification before publication:

- 1) Where do the authors place the Leucokinin neurons that have earlier (Yurgel et al 2019 Plos Biol) been shown to modulate feeding state?
- 2) Allatostatin neurons are thought to inhibit feeding in adult *Drosophila* (Hergraden et al., PNAS 2012). However, activation of AstA neurons by Chr does not appear to change food intake in the assay shown here. This needs to be addressed in the context of how AstA helps resolve sensory conflict during feeding.
- 3) The change in preference by activating and inhibiting AstA neurons is remarkable. Did they try expressing AstA RNAi with the activity inhibiting transgene (Gt)? Do they expect the

shift towards bitter-sweet to be lost as well? And would overexpression of AstAR on FB neurons shift food preference?

4) Several of the neuropeptide GAL4s change behaviour by either activation or inhibition but NOT both. Is it possible that these NPs help recognise the valence of a taste stimulus i.e DH44 helps recognise bitter taste? This needs to be explained better for all the identified NPs, where the shift in behaviour is with either activation or inhibition but NOT both (Figure 2).

5) The activation of DA subsets shifted preference towards bitter-sweet but activation/inhibition of the MB regions to which these neurons project had no effect on the choice – this contradictory result needs to be explained. At least some of these DA subsets project to other regions of the brain – e.g The PPL1-g2a'1 DANs project to the PAM neurons (Felsenberg et al., 2017), as well as the fan shaped body (FSB) and the lateral accessory lobe (LAL; Scaplen et al., 2020). Could these alternate projections be relevant in the context of choice?

6) Activation of TRH GAL4 leads to no feeding. Is this expected? The implication of this result needs explaining.

7) In naïve hungry flies bitter-sweet ingestion correlates with inhibition of FB neurons whereas sweet had no effect. Given that hungry flies eat both types of food equally when hungry, this result needs better explanation.

8) Fig 4f leaves out the various classes of DA neurons identified in Fig 2 – are they equivalent in their connection to the NP neurons? This is not clear and should be addressed.

9) The changes in FB neuron activity and their correlation with a choice suggests that FB neurons might have high basal activity. Is this the case and is their basal activity different in a hungry and fed fly? It would be helpful if basal activity measures of FB neurons are shown and mentioned clearly.

10) When PPL1- γ 2 α '1 DANs are inhibited, starved flies take much longer to identify a food source (Tsao et al., eLife, 2018; Sharma and Hasan, eLife, 2020). Do the authors think that the decision not to search for food in a hungry fly (in the absence of a choice) also requires activity in FB neurons? Or is the role of FB neurons only when there is a choice of food?

This aspect should be included in lines 325-326 of the discussion.

11) Finally, the authors mention how integration of upstream inputs at the FB very likely determines foraging in natural conditions. Under natural conditions flies forage by walking and by flying. Thus, any neurons that integrate information for making a foraging decision would also need to integrate sensory information from flight promoting circuits (Sharma and Hasan eLife, 2020) and this aspect should be included in the discussion.

Gaiti Hasan

Reviewer #3 (Remarks to the Author):

In this study, Sareen et al. suggest that they have identified a subset of fan-shaped body (FB) neurons in the fly brain that integrates taste quality, previous experience, and hunger state to encode behavioral decisions. They devised a food choice assay, allowing flies to choose between sweet-only and bittersweet food. They show that flies exhibited no preference between sweet-only and bittersweet food if the sucrose concentration in the bittersweet food was ten times higher than that in the sweet-only food. Under these equal-preference conditions, the authors searched for neurons whose activation or inhibition shifted the preference towards sweet-only or bittersweet food based on the idea that such neurons are likely involved in the decision-making process during flies' food choices. They tested various neurons previously shown to play a role in learning, memory, decision-

making, navigation, and hunger-driven behavior. Their positive hits represent several types of feeding- or hunger-related peptidergic neurons (including those expressing LK, AstA, NPF, or DH44), three mushroom body-innervating dopaminergic neurons (PPL1- γ 2 α '1, PPL1- α 3, and PAM- α 1), and a small subset of FB layer 6 (FBI6) neurons. Using RNAi against the receptors for the neuropeptides and dopamine, the authors mapped the potential connectivity between these neurons. Their data suggest that FBI6 neurons receives direct inputs of AstA, DH44, and Lkr and thereby acts as an integration node for these three neuropeptide signals. Finally, the authors have presented a curious scenario whereby the Ca²⁺ response in FBI6 neurons to food taste can predict the food choice made earlier by the flies. Presenting food that had previously been rejected by the fly to its foreleg caused a decrease in Ca²⁺ signal in those FBI6 neurons. Therefore, the authors conclude that FBI6 encodes "rejected choice."

Overall, the experiments presented in this manuscript have been done methodically and the presented data is of high quality. The phenomenon they have observed in this study is interesting and warrants further investigation. However, I have serious concerns about the authors' data interpretation and am not convinced that their results support the conclusions they make in this manuscript.

My first concern is their food choice assay in which flies chose bittersweet food (500 mM sucrose + 1 mM quinine) over sweet-only food containing < 50 mM sucrose. However, when the sucrose concentration in the sweet-only food was > 50 mM, more flies chose it. The authors have interpreted this finding as indicating that "the caloric advantage in choosing a less palatable bittersweet food is outweighed by the danger-avoidance advantage of the sweet option." However, no evidence is presented that the bittersweet food is less palatable than the sweet-only food when the sucrose concentration in the sweet-only food is low. Isn't it equally possible that the 500 mM sucrose masks the bitterness of quinine at the sensory level so that the bittersweet food is more or equally palatable to the sweet-only food of low sucrose concentration? Furthermore, the authors suggest that the hungeriness of the fly regulates this food choice behavior and that hungrier flies exhibit a stronger preference for the bittersweet food (L136-137). However, their results in Fig. 1b do not support this claim, showing no shift in food preference when flies had been starved for longer. The authors use this assay to probe a complex decision-making process involving value tradeoff, internal states, sensory inputs, and past experiences, but they need to provide more evidence supporting its appropriateness.

The authors show that silencing of DH44- or Lk-expressing neurons shifted flies' food preference towards bittersweet food, whereas activation of NPF neurons shifted the preference towards sweet-only food. Even if we tentatively accept that preferring bittersweet food indicates a higher hunger level, these results do not agree with several previous studies. NPF is the fly homolog of mammalian neuropeptide Y (NPY). Both NPF and NPY have been shown to be hunger signals that promote feeding and food-seeking behavior (Wu et al., *Nat. Neurosci.* 2005 & *PNAS* 2005; Inagaki et al., *Neuron* 2014; Bhagyashree et al., *Nat. Neurosci.* 2019; Beshel & Zhong, *J. Neurosci.* 2013; Krashes et al., *Cell* 2009; Tsao et al., *eLife* 2018). Therefore, activation of NPF neurons should shift flies' food preference to the bittersweet rather than sweet-only option. Similarly, DH44 neurons are activated by nutritious sugar to promote food intake (Dus et al., *Neuron* 2015). Accordingly, silencing DH44 neurons is expected to make flies favor sweet-only food. Finally, two recent studies (Yurgel et al., *PLoS Bio.* 2019; Bhagyashree et al., *Nat. Neurosci.* 2019) have shown that some LK neurons are activated by starvation to drive feeding and food-seeking behavior. Activation of LK neurons also decreases postprandial sleep (Murphy et al., *eLife* 2016).

These studies suggest that LK is a hunger signal, which is inconsistent with the findings of the current manuscript implying increased perceived hunger upon inhibition of LK neurons. Consequently, I feel it is important for the authors to compare and discuss their findings with those previous studies.

The authors have studied the role of FBI6 neurons by mainly using c205-GAL4 in their behavioral experiments. The authors do not show its expression pattern, but c205-GAL4 also strongly labels a group of neurons in the subesophageal zone (SEZ), which is a convergence site for gustatory inputs (Hu et al., Cell Reports 2018). The authors should validate all their c205a-GAL4 results using 84C10-GAL4. They do present some behavioral experiments using 84C10-GAL4 in Extended data fig. 3, but they are not mentioned in the main text. The baseline preference in Extended data fig. 3b appears to be different from those shown in the main figures. Moreover, there is no control group for the data in Extended data fig. 3d. The authors should consider performing these experiments again using more consistent conditions and appropriate controls. There is a strong possibility that neurons in the SEZ are regulated by hunger signals to control feeding behavior. Therefore, it is also critical that the authors verify their receptor knockdown in the FBI6 experiments using 84C10-GAL4.

Finally, I have many questions regarding their calcium imaging experiments. First, it is not clear how the experiments were performed. Were flies immediately imaged after the choice assay or was there a waiting period? Were the flies that underwent the choice assay food-deprived? If yes, for how long? I feel the authors need to describe their experiments in more detail.

Second, if inhibition of FBI6 neurons represents “rejected choice”, why does bittersweet and not sweet-only taste inhibit FBI6 neurons? Aren't these two foods in an equilibrium condition, i.e., flies exhibit no preference for either?

Third, if I understand the notion correctly, the authors have assumed that when a fly chooses one food option, it will choose it again the next time. This assumption may not be correct and requires further investigation. It is known that behavioral expression is probabilistic. For example, flies that choose incorrectly in a memory test display a likelihood of choosing correctly in a retest equal to that of flies who made the correct choice in the initial test (Cervantes-Sandoval & Davis, Curr. Biol. 2012). The authors should validate if flies indeed choose the same option repeatedly in their choice assay.

Fourth, if inhibition of FBI6 neurons represents “rejected choice,” why does silencing FBI6 neurons shift the food preference towards the bittersweet option? The authors' explanation for this scenario is that it is a default behavioral decision (L322-324), which is not satisfactory. I suggest the authors further test their model by (1) giving flies a choice between two areas containing identical food and illuminating one area with green light to inhibit FBI6 neurons via GtACR1. Based on their model, the flies should reject the lit area. (2) Give the flies a food choice by which they will clearly display a preference for one food type over the other. Then activate FBI6 neurons via CsChrimson and establish if this treatment compromises the flies' decision-making and reduces their preference.

Fifth, the argument that hungrier flies are also thirstier (L254-256) is unsubstantiated. The authors must directly test this hypothesis by measuring both food and water intake.

REVIEWER #1

In this manuscript, Sareen, McCurdy, and Nitabach seek to identify neurons and neuromodulatory systems involved in the integration of external food cues and internal states to perform adaptive feeding decisions. They first develop a two-choice feeding assay in which flies can choose to eat from two sides of a plate coated with different tastants (bittersweet vs various concentrations of sweet). They identified an ensemble of cells types – notably, Leucikinin neurons, Allatostatin neurons, NPF neurons, DH44 neurons, and aversive PPL dopamine neurons that influence feeding decisions and feeding amount. They also identify a set of layer 6 fan-shaped body neurons whose silencing biases the flies towards eating from the bittersweet area. The authors perform a very nice set of RNA knock-down experiments in these cells to show that the effects of manipulating peptidergic and neuromodulatory neurons are consistent with the effects of manipulating receptors for these signaling molecules in this downstream neuron. Finally, the authors use calcium imaging to show that the activity of L6 FB cells changes with feeding state, and that responses to the two tastants used in their study vary with which food the animal ate in the feeding assay. On the basis of these experiments, the authors claim that L6 FB neurons encode behavioral choice and integrate taste quality, choice outcome, and hunger state to influence behavioral decisions. These results are potentially novel and interesting, however there is an important confound that the authors need to address to support their claims.

We thank Reviewer #1 for their insightful comments and suggestions for improvement. We have performed their suggested experiments, added requested discussion, and cited the suggested literature. Please see our point-by-point response below.

Major issues:

1) Imaging experiments do not distinguish between choice encoding and sensory adaptation.

A key claim of this paper is that the activity of FBL6 neurons reflects feeding decision of the fly. This conclusion was made because FBL6 neurons are inhibited when flies experience a food type that they previously rejected (sweet or bittersweet), or that they innately reject (bittersweet in naïve flies). An alternative explanation for this observation is that FBL6 neurons, or the circuitry upstream of FBL6, undergoes adaptation as a result of prolonged food exposure. In the choice paradigm used in this study, the flies likely experience/sense the ‘chosen’ food more than the ‘non-chosen’ food. If true, then unequal food exposure could cause greater adaptation towards the chosen food over the non-chosen food, and a result, lead to different responses in the FBL6 neurons. To rule out adaptation as the reason for differential encoding of the “chosen” food, the authors should consider experiments putting flies on a purely sweet circular arena or a purely bittersweet circular arena for the same amount of time as in the choice assay, and then measuring FBL6 neural activity through Ca imaging. If responses are similar as in the choice assay, it would suggest that the differential encoding reflects simple sensory adaptation. Alternatively, if the differential encoding is not observed here, it would provide stronger evidence that the activity in FBL6 reflects a “decision” or “choice”.

This is a conceivable alternative interpretation of the FBL6 neural responses we have observed. To address this, we have performed additional experiments suggested by this reviewer. We have

also clarified in the manuscript the relevance of already-performed experiments to this interpretation. These existing and new experiments indicate that the differences in FB16 neural response we observe are not due to sensory adaptation. In addition, we have refined our model in light of these new experiments and interpretations.

1. We performed the experiments suggested by this reviewer: exposed food deprived flies to only one type of food in the behavior assay (“only sweet”, **Suppl. Fig. 4a-b** or “only bittersweet”, **Suppl. Fig. 4c-d**), and measured FB16 neural activity in response to both sweet and bittersweet stimuli (**Suppl. Fig. 4a-d**, new). As shown in these new figures, FB16 responses to sweet or bittersweet food stimuli are completely unaffected by prior exposure to these food stimuli in the “no choice” context of behavioral assay arenas containing only one food stimulus. This rules out the interpretation that sensory adaptation (or other purely sensory mediated phenomena) underlies different FB16 responses in flies that choose different food stimuli for consumption. Rather these new results reinforce the conclusion that active choice is the key determinant of FB16 response to sweet and bittersweet stimuli. The new results are described in the Results section in Line 303-316.
2. We previously showed in **Fig. 4d** that FB16 neural activity is strongly inhibited by only bittersweet stimulus and not sweet stimulus in naïve food-deprived flies (**Fig. 4d** “naïve deprived”). Since these “naïve deprived” flies were never exposed to either food stimulus prior to imaging, this difference in FB16 activity in response to sweet and bittersweet stimuli can’t be explained by sensory adaptation.
3. In addition, almost all flies that experienced the conflicting food choice assay exposed themselves to both foods by walking on both sides of the assay arena, as determined by automated video tracking (**Suppl. Fig. 1g-j**). There were flies in this group that chose not to consume either food option (**Fig. 4d** “chose neither”). These flies that chose to eat no food exhibited strong FB16 inhibitory response to both food stimuli and no attenuation in FB16 response despite ample sensory exposure of the forelegs to both food options. These data also indicate that sensory adaptation of FB16 neural responses during exploration fails to account for subsequent FB16 response. Rather, as above, these results reinforce the conclusion that active choice is the key determinant of FB16 response to sweet and bittersweet stimuli.

New text in Line 303-316:

To further test whether FB16 neural response to the previously tested sweet and bittersweet taste sensory cues is absolute or context-dependent, we presented food-deprived flies with a no-choice assay in which only one type of food (either sweet or bittersweet) was present. We then measured FB16 neural responses to both sweet and bittersweet taste stimuli in the flies that consumed the only available food option in this no-choice context (Supplementary Fig. 4a-d). FB16 neurons were inhibited by neither the taste of food that flies experienced and consumed during the no-choice task, nor by the new taste stimulus that flies did not experience before (Supplementary Fig. 4a-d). Since these flies experienced food in a no-choice context, they had a different prior experience

from the flies that experienced the conflicting two-choice task, which modified their FBL6 neural response to these stimuli. Flies that consumed the only food option available to them showed no response to the consumed taste stimulus. Thus, it is possible that lack of response to a novel food taste reflects acceptance of that novel food option. In summary, strong inhibitory FBL6 neural responses to taste stimuli are context-dependent and correspond to rejected food choice, suggesting that context-dependent temporally coupled inhibitory FBL6 responses are the neural representation of rejected behavioral food choice.

2). A conceptual model linking the observed FBL6 activity to behavioral decisions is not presented.

The authors identified an ensemble of neurons that converge onto FBL6 cells to influence feeding decisions; however, they do not provide hypotheses about how these neuromodulators might generate the inhibition (or lack of inhibition) observed in FBL6 neurons under different conditions. It's also unclear how inhibition or lack of inhibition in FBL6 neurons might influence feeding behaviors.

We thank this reviewer for highlighting the importance of elaborating and clearly stating our model and hypotheses. We have addressed these suggestions in detail below and also added a revised schematic in **Fig. 5f** (new).

A paragraph in the discussion, and perhaps a figure suggesting (1) how the various neuromodulators and neuropeptides might affect the physiology of FBL6, and

We previously discussed how receptor RNAi knockdown of neuromodulators secreted by upstream neuromodulatory neurons may be connected to downstream FBL6 neurons in the Results section. Based on our findings, we have now added to this discussion our hypotheses about how upstream neuromodulatory neurons might affect the physiology of FBL6 neurons. Our data suggest that inhibition of AstA, DH44, and Lk neurons will result in inhibition of FBL6 neurons. In addition, AstA-R, DH44-R, and Lkr are all GPCRs that potentially couple to different G-proteins in FBL6 neurons activating different signaling pathways to transduce signals encoded by these neuropeptidergic neurons for integration. Based on our data, dopaminergic neurons modulate neuropeptidergic neurons, which send signals to FBL6 neurons.

New text in Line 209-222:

Since inhibition of AstA, DH44, and Lk neurons is equivalent to receptor RNAi of these neuropeptides in target neurons such as FBL6, and both of these manipulations produce the same shift in preference for bittersweet as optogenetic inhibition of FBL6 neurons alone, these findings strongly suggest that inhibition of AstA, DH44, and Lk neurons results in inhibition of FBL6 neurons. The neuropeptidergic receptors for AstA, DH44, and Lk are GPCRs that potentially couple to distinct G proteins in FBL6 neurons that interact with different effector molecules to produce distinct downstream cellular responses. For example, AstA-R1 human homolog galanin receptor is coupled to Gi and galanin binding initiates different intracellular signaling pathways in different target

tissues⁶⁷. *DH44-R1* belongs to CRF-like secretin family of GPCRs that stimulate adenylylate cyclase and can activate multiple $G\alpha$ subunits⁶⁸, and *Drosophila Lkr* couples to Gq signaling pathways^{69, 70}. These distinct G protein coupled downstream signaling pathways can maintain identities of signals encoded by the neuropeptides for integration in *FBL6* neurons by coupling to ion channels, neurotransmitter receptors, and other biophysical and biochemical effectors to differentially modulate cellular excitability and synaptic transmission.

(2) how *FBL6* activity might influence behavioral decision through the fan-shaped body would help clarify the authors' hypotheses and suggestions for future experiments.

We thank this reviewer for making us think harder about our conceptual model and further refine it. We previously argued that persistent, temporally and spatially indiscriminate, optogenetic inhibition of *FBL6* throughout the decision assay disrupts the integration of signals that *FBL6* receives for forming a decision. We have now performed further behavior experiments to test our model by spatially limiting optogenetic *FBL6* inhibition to one side of the arena while presenting the same sweet food on both sides (**Fig. 5a**, new). Control flies not expressing *GtACR1* avoid food on the lit side of the arena, indicating intrinsic aversion to bright green light used in the assay. This finding is consistent with previously published work showing that flies prefer to eat from dimly lit areas compared to brightly lit areas (Rieger et al., *J Biol Rhythms* 2007). In stark contrast, this intrinsic aversion is abolished in flies expressing *GtACR1* in *FBL6* neurons (**Fig. 5a**, new), which exhibit equal preference for the lit and dark sides. This indicates that optogenetic inhibition of *FBL6* in this “no choice” context induces a preference for the lit side of the arena that counteracts intrinsic green light aversion.

This strong effect of optogenetic inhibition of *FBL6* activity in this additional, but different, food-related “no choice” context reinforces *FBL6*'s key role in food-related decision making. These results again demonstrate that imposing an unnatural inhibitory activity pattern on *FBL6* neurons during decision making that is not driven by specific sensory inputs perturbs the decision making process. In these new experiments, optogenetic inhibition of *FBL6* is spatially, yet not temporally, confined, unlike our prior experiments with spatially homogenous optogenetic inhibition throughout the food arena (**Fig. 3a**, left panel, **Suppl. Fig. 3b** left panel). Interestingly, these effects are the opposite of what would be expected from a simplistic ON/OFF sensorimotor program in *FBL6* that turns feeding on and off on command, independent of the context of sensory environment and internal state at any given moment. Inspired by these new results, we have refined our model to reflect transient and dynamic processing of sensory and internal state information and encoding of choice in these *FBL6* neurons.

Based on the imaging experiment suggested by the reviewer in comment 1 (**Suppl. Fig. 4a-d**, new) and new behavioral data in **Fig. 5a**, we have now adapted a “synaptic gating” model used in the context of mammalian circuits gating flow of information to the cortex (see below) to explain our results. As suggested, we have also added a revised schematic in **Fig. 5f**, which replaces the prior **Fig. 4g** schematic and highlights the points suggested by the reviewer. These new behavior results and how they inform our model are discussed below:

New text in Line 317-353:

FBI6 neurons are strongly inhibited by a single stimulation with rejected food choice (Fig. 4c-d), this inhibition is context-dependent, and persistent sensory-stimulus independent optogenetic inhibition of FBI6 neurons throughout the decision assay shifts food preference to bittersweet (Fig. 3a left panel, Supplementary Fig. 3b left panel). We thus hypothesized that imposing unnatural, temporally and spatially homogenous optogenetic inhibition of FBI6 throughout the decision task disrupts value integration and decision making. This imposed activity pattern, that does not take into account the sensory environment and current value estimates of the fly in any given moment, results in the hungry fly defaulting to higher calorie bittersweet food. We tested this model further by optogenetically inhibiting FBI6 neurons on only one side of the arena while presenting flies with the same sweet food on both sides. We found that control flies not expressing GtACR1 avoided the lit side of the arena indicating intrinsic aversion to bright green light⁷⁵. In contrast, persistent FBI6 optogenetic inhibition mediated by GtACR1 abolished this bright green light avoidance, resulting in equal preference for food on both sides (Fig. 5a, Supplementary Table 1). This indicates that persistent optogenetic inhibition of FBI6 induces a positive preference that counteracts the intrinsic aversion to bright green light. While spatially and temporally homogeneous optogenetic inhibition of FBI6 neurons during decision making shifted preference to bittersweet (Fig. 3a left panel, Supplementary Fig. 3b left panel), spatially restricted optogenetic inhibition shifted preference to sweet (Fig. 5a). Congruent with context-dependent FBI6 neural response to taste stimuli (Fig. 4c-d, Supplementary Fig. 4a-d), optogenetically inhibiting the same FBI6 neurons in different behavioral contexts produced different choice outcomes.

These findings are inconsistent with predictions of a simplistic ON/OFF sensorimotor model in which FBl6 neurons would turn feeding on and off on command independent of context. Instead these findings support a model in which the activity of FBl6 neurons is transient, dynamic, and highly state- and context-dependent. We hypothesize that, similar to mammalian nucleus accumbens and other cortical circuits involved in gating the flow of information to cortex for attention and decision-making^{76, 77, 78}, FBl6 neurons are bistable and oscillate between two states that are characterized by different values of membrane potential. According to the synaptic gating model^{76, 77, 78}, in the “down state”, FBl6 neurons would be hyperpolarized (i.e., closed gate) leading to rejection of an option. To reach the “up state”, i.e., a membrane potential just below the neuron’s firing threshold, FBl6 neurons would have to receive input from another neuron. Only when FBl6 neurons are in this “up state” (i.e., open gate) can subsequent inputs trigger action potentials leading to acceptance of an option influenced by these inputs. Similar to mammalian cortical gatekeeper neurons^{76, 77, 78}, FBl6 neurons could act as a gatekeeper that integrates converging inputs from upstream neuropeptidergic neurons using non-linear summation (Fig. 5f). If the activity of FBl6 neurons were forced into an unnatural “down state” using optogenetic inhibition, the integration of inputs to FBl6 neurons would be disrupted. Under such a condition, hardwired innate inputs could be shunted to bypass the gatekeeper circuit motif resulting in unexpected behavioral outcomes that would not take into account adaptive flexible input variables such as the context in which flies choose food options.

New text in Line 389-431:

Animals assess and assign value estimates to internal state and external environmental parameters before integrating these estimates to guide adaptive decision making. Upstream neuromodulatory subsets in the decision ensemble we identified have known roles in hunger dependent food intake^{7, 10, 12, 83}, reward⁸⁴, valence^{22, 62}, and long-term memory^{18, 22, 62}. Since under natural foraging conditions, flies conduct food search while walking and flying, it is expected that sensory information about food seeking during flight would also be integrated in this ensemble. Consistently, we also identified flight-promoting dopaminergic neurons involved in food seeking behaviors^{31, 63}. These modulatory neurons are well positioned to estimate value of features in the sensory environment and internal state that can be updated in a state- and context-dependent manner. For example, AstA neuron activity influences relative carbohydrate and protein preference⁷, while DH44 neurons sense sugars⁸³ and amino acids¹². Thus, AstA and DH44 neurons could convey food identity information to FBl6. NPF neuron activation is inherently rewarding⁸⁴, and thus could convey food valence information. Lk neurons have been implicated in nutrient sensing^{10, 81}, and their activity regulates feeding in food deprived flies (Fig. 2a-b), suggesting that internal metabolic state information could reach FBl6 through Lk/Lkr signaling. FBl6 neurons project their axons into the fan-shaped body^{53, 54}, have dense dendritic projections in the superior medial protocerebrum (SMP), and have sparse dendritic projections in superior intermediate protocerebrum (SIP) and superior lateral protocerebrum (SLP)^{53, 54} (Fig. 5c). AstA neurons form postsynaptic connections with FBl6 neurons in the SMP and SIP regions (Fig. 5d-e), and Lk^{80, 81} and DH44^{11, 82} neurons have extensive projections in the SMP, SLP, and SIP

regions. It is probable that FBl6 neurons receive decision relevant signals from these upstream neuropeptidergic neurons in these higher brain regions (Fig. 5f). Dopaminergic subsets involved in aversive memory and modulation of flight and feeding (PPL1 $\gamma 2\alpha'1$)^{62, 63}, taste conditioning (PPL1 $\alpha 3$)²², and long-term memory (PAM $\alpha 1$)¹⁸ also affect food choice in the decision assay (Fig. 2a), and could provide signals for predicting and updating value estimates in working memory, analogously to primate dopaminergic ventral tegmental area⁸⁵. Since the identified dopaminergic neurons project to SMP, SLP, and SIP regions, where the neuropeptidergic neurons also arborize and connect to FBl6 neurons, it is conceivable that it is in these same higher brain regions that the identified dopaminergic neurons directly modulate neuropeptidergic neurons (Fig. 2b-e) to update value estimates encoded by them.

As the value estimates are updated in a state- and context-dependent manner, they are dynamically received by FBl6 neurons. Similar to mammalian cortical gatekeeper neurons^{76, 77, 78}, FBl6 neurons could be bistable, transitioning between a hyperpolarized “down state” and a resting state associated “up state.” According to such a synaptic gating model, when FBl6 neurons are in “down state,” choice outcome is rejection. When FBl6 neurons receive inputs to reach “up state” with membrane potential just under firing threshold, subsequent inputs are integrated non-linearly. If this non-linear summation crosses the threshold to depolarize FBl6 neurons, choice outcome is acceptance. FBl6 neurons could act as a gatekeeper circuit motif that non-linearly integrates converging inputs and gates the flow of information to downstream circuits to initiate motor programs implementing the decision (Fig. 5f). This transient dynamic representation of choice is updated as the value estimates change over time with animal’s experience, internal state, and sensory environment. Such transient dynamic neural activity representing a constantly updated choice would require extensive recurrent circuitry between the inputs and outputs of the decision ensemble. Congruently, SMP, SLP, SIP, and FB brain regions have extensive recurrent circuits⁴³ that may be involved in such dynamic updating and integration for decision making.

3) The treatment of hunger and encoding of “internal state” is confusing. It is true that 84C10 neurons show different responses when starved or fed, but the main behavioral assay used in the paper does not shift its equilibrium point with hunger state and c205 manipulations do not alter feeding based on hunger state.

We agree that our explanation and analyses of hunger effects on food choice were not stated as clearly as possible. We have now clarified this with edits to the text, new data, and new statistical analyses.

1. Our data in **Fig. 1b** suggest that equal preference for sweet and bittersweet is extended from the 50 mM condition to the 100 mM condition when flies are food deprived for 21 h (**Fig. 1b, Suppl. Fig. 1d**). To determine if this apparent shift in preference from sweet to equal preference significantly varied with food deprivation duration, i.e., preference shifted from sweet towards bittersweet with hunger level, we performed one-way ANOVA followed by post-hoc test for linear effect in the data. We found that the slope of preference index over deprivation time was negative and this slope was statistically different from zero (**Suppl. Fig. 1d**). Statistically significant linear effect in these data

with a negative slope indicates decreasing preference for sweet as flies are food deprived from 2 h to 21 h (**Suppl. Fig. 1d**, slope = -0.165, $p = 0.0149$).

2. We have also provided new additional data in **Supplementary Fig. 1e-f** from wild-type strain Canton S, (CS). Wild-type CS flies prefer sweet at the 50 mM sucrose vs. 500 mM sucrose + 1 mM quinine condition when food deprived for only 5h compared to equal preference for sweet and bittersweet when food deprived for 21h. This difference in preference is statistically significant (**Suppl. Fig. 1f**, unpaired t-test $p = 0.016$). These new data, in a wild-type strain distinct from the *w1118* control strain required for genetic experiments, establish that the equal-preference point does change with hunger state. Note that ANOVA with post-hoc linear effect analysis or t-test are employed in these two contexts as appropriate for the number of durations of food deprivation being compared.

Together, these data demonstrate that equilibrium point for food preference shifts with hunger state.

The authors should clarify their language describing the circuits they found as encoding or integrating “internal state” to inform a decision. See several points below.

As suggested, we have explicitly defined our terminology of “encoding” versus “integration” of internal hunger state to clarify our interpretation and conclusions.

New text in Line 172-178:

An animal’s internal state is a complex amalgamation of its current sensory and internal environment, such as hunger level, thirst level, motivation to feed, sleep or mate etc. To facilitate description and interpretation of our results in this study we define “encoding” and “integration” as follows. By “encoding” of hunger state we mean the estimation and representation of hunger level by specific neurons. By “integration” we mean the combination of hunger state with other internal state and external sensory information to drive food choice behavior.

Previously, we presented data that showed that optogenetic activation or inhibition of FB16 neurons did not halt or initiate feeding (**Fig. 3b** and **Suppl. Fig. 3c**), nor changed the amount of food consumed by flies (**Fig. 3c-d**). These data excluded the possibility that FB16 neurons encode hunger state per se (as we define it). Instead, we hypothesized that Lk neurons encode hunger state since their activation abolishes feeding in food deprived flies (**Fig 2b**, **Suppl. Fig. 2b**, and Line 400-402). Based on these results, we hypothesized that FB16 neurons integrate internal hunger state information, which they receive through Lk, with other information such as food valence to form a decision. As suggested, we have now clearly stated our definitions of encoding and integration to clarify our interpretation and hypotheses.

Additional comments:

The majority of the behavior was performed in a different Gal4 line than the imaging. I did

not see a reference to Extended data Fig. 3 in the text (?) and so completely missed this important control on my first read. This needs to be prominently cited and clarified.

We have now emphasized this important control and described 84C10-GAL4 data from **Suppl. Fig. 3** in the main text. Additionally, we have performed new experiments confirming receptor RNAi results using 84C10-GAL4 (**Suppl. Fig. 3b** right panel). In an important additional new control experiment, we have now also shown that deletion of FBI6 expression from the c205 expression pattern using 84C10-LexA and LexAop-GAL80 abolishes the shift in food preference induced by optogenetic inhibition using c205 on its own (**Fig. 3a**, “84C10>GAL80+c205>Gt”).

The section describing FBL6 activity in response to water was confusing and under-developed. Several claims are made about these responses which rest on speculation. The authors might either investigate their thirst hypothesis further or leave this section out.

We agree that the section on water responses in FBI6 requires further investigation. Since these results are not germane to the key conclusions of this manuscript and merit detailed inquiry beyond the scope of this study, as suggested, we have removed this section.

Although it is not necessary for the current manuscript, imaging from some of the upstream neuromodulatory neurons identified in this manuscript would clarify where the signals observed in L6FB emerge and what types of signal transformations occur between these populations.

We absolutely agree with this, and we are extremely interested in future investigation of how the signals from upstream neuromodulatory neurons are processed, integrated, and transformed in FBI6 under different internal states and contexts. While we intend to conduct these experiments in the future, we also agree with the reviewer that this important and exhaustive investigation is not necessary for the current manuscript and beyond the scope of the current study.

Line 76: “This equal-preference point was identical at all of the tested food deprivation durations, suggesting an external taste sensation and internal hunger state equilibrium at this concentration ratio.”: Can you claim that the external taste sensations are in balance with internal hunger state if manipulating the hunger state does not alter the equal-preference point? The data suggests that the equal-preference point is invariant to internal hunger state and only depends on a balance of external sensory cues. See point (3) above.

We have provided new statistical analysis for *w1118* (**Suppl. Fig. 1d**) and new data from CS wild-type strain (**Suppl. Fig. 1e-f**) demonstrating that the equal-preference point does change with hunger state (see detailed response to point (3) above).

We agree that our previous language was confusing and have refined our language to clarify these results.

New text in Line 63-70:

At 10- and 5-fold sucrose concentration ratios (Fig. 1b, 50 mM vs. 500 mM sucrose + 1 mM quinine, 100 mM sucrose vs. 500 mM + 1 mM quinine at 21 h) almost all flies consumed only a single food option with half of them consuming only sweet and half consuming only bittersweet resulting in equal-preference for sweet and bittersweet at these conditions. The equal-preference point varied with food-deprivation duration, shifting preference from sweet toward bittersweet with increasing food-deprivation duration (Fig. 1b, Supplementary Fig. 1d-f, Supplementary Table 1), suggesting an external taste sensation and internal hunger state equilibrium at the equal-preference concentration ratios.

Line 100: “While neural substrates integrating internal state and external stimuli in the *Drosophila* brain remain unclear, there are hints in the literature of cell populations and brain regions that could be involved.” I don’t think this is a fair statement. Numerous papers have explicitly examined this question in different behavioral contexts. For example, Sayin et al. 2018 explicitly identifies several MBONs, DANs, and octopaminergic neurons as encoding and integrating odor and taste stimuli to dynamically shape the intensity of odor tracking behavior in a hunger-dependant manner. Work from the Wang lab (Root and Wang 2011, Ko and Wang 2015) have identified molecular mechanisms at the olfactory periphery that allow hunger state to shape odor encoding. Vogt and Samuel 2020 show that in *Drosophila* larvae, hunger modulates a serotonergic neuron that alters the routing of odor information through the antennal lobe to influence behavioral decisions. In the courtship system, various neurons integrate internal state information to shape the intensity of courtship behaviors (Deutsch and Murthy 2020, Hindmarsh and Ruta 2020). These studies should be cited here.

We did not intend to imply that candidate neural populations for integration of internal state and external stimuli have not been previously identified, and thank the reviewer for pointing this out. To clarify that molecular and biophysical mechanisms of such integration are poorly understood, while highlighting that cell populations have been identified in which such integration might occur, we have modified our language and cited the suggested papers,

New text in Line 90-96:

*While the mechanistic basis of integration of internal state and external stimuli on a molecular or biophysical level in the *Drosophila* higher-order brain remains unclear, prior studies reveal some cell-populations and brain regions that are involved in various decision making contexts. Various neuromodulatory and other neurons regulate hunger dependent food intake^{7, 8, 9, 10, 11, 12, 13}, hunger dependent odor encoding and food search^{14, 15, 16}, reward^{17, 18, 19, 20} or punishment²¹, memory^{18, 20, 22}, and internal state for courtship^{23, 24, 25}. The mushroom body is an insect central brain region involved in gustatory learning and memory^{26, 27} and valence encoding²⁸, and is thought to be a major center controlling higher-order behaviors¹⁴, including associative learning^{29, 30, 31}.*

Line 137: “Feeding suppression induced by Lk neuron activation implies a decrease in perceived hunger of food-deprived flies, consistent with the shift towards higher calorie bittersweet food by Lk neuron inhibition reflecting increased perceived hunger.” This logic does not seem clear. Hunger itself does not shift the animals’ preference for sweet vs bittersweet. See point (3) above.

We have now provided new additional data and analysis of our previously presented results showing that hunger itself does indeed change the fly's preference for sweet vs. bittersweet (also discussed above for point (3), **Suppl. Fig. 1d-f**). Based on these results, we hypothesize that since optogenetic activation of Lk neurons suppresses feeding in food deprived flies (implying satiety), optogenetic inhibition of Lk neurons could have the opposite effect, i.e., increased hunger.

Line 208: “Taken together these results demonstrate the FBI6 does not encode or affect metabolic signals of hunger or satiety” seems in conflict with line 218: “We hypothesized that FBI6 integrates internal hunger state and external food-related value estimates with experiential information to drive decisions”

This point is related to the reviewer's point (3) above about the distinction between encoding and integration of internal hunger state. As detailed above, we have now provided explicit definitions of encoding and integration to clarify our interpretations and conclusions.

New text in Line 172-178:

An animal's internal state is a complex amalgamation of its current sensory and internal environment, such as hunger level, thirst level, motivation to feed, sleep or mate etc. To facilitate description and interpretation of our results in this study we define “encoding” and “integration” as follows. By “encoding” of hunger state we mean the estimation and representation of hunger level by specific neurons. By “integration” we mean the combination of hunger state with other internal state and external sensory information to drive food choice behavior.

Based on these definitions, FBI6 neurons integrate hunger state information, which we hypothesize is encoded by Lk neurons, with other external and internal information.

Line 198: “FBI6 activity manipulation neither prevented food consumption by food-deprived flies nor induced food consumption by fed flies”. This finding is inconsistent with the claim made elsewhere that FBI6 integrates internal and external cues to generate a behavioral decision. (Related to point 3 above)

Based on the explicit definitions of encoding and integration we have added (see also above and Line 172-178), there is no inconsistency in the interpretation that FBI6 does not encode hunger level but rather integrates the hunger state signal (that it receives from Lk) with other signals, such as food valence, to control food-related decisions.

Line 225: “FBI6 neurons of naïve food-deprived flies were strongly inhibited by bittersweet but not sweet.” It looks like there is some response to the sweet stimulus in this plot.

While there may appear to be a subtle inhibitory response to the sweet stimulus, this response is statistically insignificant, in contrast to the very strong statistically significant inhibitory

bittersweet response (**Fig. 4d**, Kruskal-Wallis statistic = 37.79, $p < 0.0001$; post-hoc two-tailed Wilcoxon matched-pairs signed rank test between sweet and bittersweet response in naïve deprived flies, sum of positive, negative ranks = 47, -8, $p < 0.05$).

Line 227 “Flies often find bittersweet food aversive and thus inhibitory taste responses in FBl6 to bittersweet food may represent rejection of this option”. I don’t think that can be concluded in this experiment because the flies have not been exposed to any choice and there are many situations where these two stimuli produce a 50-50 choice in their assay.

We provide a two-part justification for this conclusion which is based on two different fly states:

1. Previous studies establish that flies innately find bitter compounds (including quinine) mixed with sugars aversive (Lee, Moon and Montell, PNAS 2009, Sellier et al., Chem Senses 2011, French et al., J Neuro 2015), which is demonstrated by reduced proboscis extension response to and consumption of bitter-sweet mixtures (Masek, Scott, PNAS 2010, French et al., J Neuro 2015, Lee, Moon and Montell, PNAS 2009, Sellier et al., Chem Senses 2011, French et al., J. Neuroscience 2015). Flies find bittersweet food aversive, as supported by these published studies, and our results show that inhibitory FBl6 response correspond with rejected choice in all other conditions that we tested (**Fig. 4c**). We thus posited that strong FBl6 inhibition by one stimulation with bittersweet taste in naïve food-deprived flies represents rejection of this stimulus. We have now added this point in the Results section.

New text in Line 270-273:

Flies innately find bittersweet food aversive²⁷, and reduce proboscis extension response to and consumption of bittersweet mixtures^{72, 73, 74}. Thus inhibitory taste responses in FBl6 to the first encounter with bittersweet food may represent rejection of this option.

2. We would like to clarify that the FBl6 neural responses shown in **Fig. 4c-d** are in response to only the first instance of a naïve fly experiencing either sweet or bittersweet stimulus. This is different from what happens in the decision assay context. In the behavior decision assay, flies are free to sample both options as many times as they choose before consuming either food. Almost all individual flies consume only either sweet or bittersweet food. This means that under the equal-preference condition approximately 50% of flies only ate sweet food while the other 50% only ate bittersweet, resulting in a preference index of zero (i.e., equal-preference). From this equal-preference condition, for imaging we selected individual flies that ate sweet only (**Fig. 4c-d**, “choose sweet”), bittersweet only (**Fig. 4c-d**, “choose bittersweet”), or no food (**Fig. 4c-d**, “choose neither”). Since these individual flies chose only one type of food, we were able to map their FBl6 neural response to their individual chosen or rejected food option.

We have now added a description of how the preference index was calculated at the beginning of the Results section to aid interpretation of results (new text in Line 52-55) and also refined our language to clarify what equal-preference means (new text in Line 63-70).

New text in Line 52-55:

Food choice was quantified by scoring the color of the food in each fly's abdomen at the end of the assay. A preference index was then calculated by subtracting the number of flies that ate bittersweet food from the number of flies that ate sweet food divided by the total number of flies that consumed food in that trial.

New text in Line 63-70:

At 10- and 5-fold sucrose concentration ratios (Fig. 1b, 50 mM vs. 500 mM sucrose + 1 mM quinine, 100 mM sucrose vs. 500 mM + 1 mM quinine at 21 h) almost all flies consumed only a single food option with half of them consuming only sweet and half consuming only bittersweet resulting in equal-preference for sweet and bittersweet at these conditions. The equal-preference point varied with food-deprivation duration, shifting preference from sweet toward bittersweet with increasing food-deprivation duration (Fig. 1b, Supplementary Fig. 1d-f, Supplementary Table 1), suggesting an external taste sensation and internal hunger state equilibrium at the equal-preference concentration ratios.

Line 236: taste responses were subsequently measured in flies that chose sweet food in the decision task. How was choice quantified here? What were the criteria for having chosen sweet or bittersweet? What was the distribution of choices?

This information was previously provided only in the Methods section. We have now added this information to the Results section to clarify the experimental design and interpretation of results.

New text in Line 284-296:

To test this, we measured FBL6 neural response to sweet and bittersweet tastes in flies that had experienced the equal-preference decision assay. Almost all flies visit both the sweet and bittersweet food sectors (Supplementary Fig. 1h) during the decision task (Supplementary Fig. 1g-i), transitioning from one sector to the other multiple times over the 5 min choice assay. These transits provide flies with sensory information about both food options via gustatory receptors on their legs, since flies are confined to stand on the food surface. The food option that an individual fly consumes is recorded by scoring the color of its abdomen at the end of the assay. Almost all individual flies consume either sweet or bittersweet food only, meaning that under the equal-preference condition approximately 50% of flies ate sweet and the other 50% ate bittersweet food resulting in an equilibrium for the population of flies within a trial. From this equal-preference trial, for imaging we selected individual flies that consumed sweet only, bittersweet only, or no food. Since these individual flies chose only one type of food, we were able to map each fly's subsequent FBL6 neural responses to its chosen and rejected food options.

Line 286-288: DH44 and AstA neurons: how direct are the connections from these to FBL6?

To our knowledge, there are no published reports of direct synaptic connections between FBL6 and Lk, DH44, or AstA neurons. To address this, we have now performed EM connectivity

analysis between AstA neurons and FBl6 neurons traced in the publicly available adult *Drosophila* Hemibrain EM dataset and found direct connections between them. Since traced Lk and DH44 neurons are not currently available in the dataset, we could not determine if Lk and DH44 neurons also form synaptic connections with FBl6 neurons. However, previously published light-level expression patterns for LK (Murphy et al., *Elife* 2016, Yurgel et al., *PLoS Biol* 2019) and DH44 (Cannell et al., *Peptides* 2016, Nassel et al., *Cell Tissue Res* 2020) show extensive arborization in the SMP, SLP, and SIP regions. Thus, we hypothesize that Lk and DH44 neurons also form direct synaptic connections with FBl6 neurons in these regions. We describe these results in new **Fig. 5b-e** and new text in Line 367-379:

To assess whether AstA, DH44, and Lk neurons form direct synaptic connections with FBl6 neurons (Fig. 5b-c), we queried the publicly available Drosophila hemibrain EM connectome dataset⁷⁹ for these neurons. We identified two traced AstA neurons both of which form postsynaptic connections with FBl6 neurons in the SMP and SIP higher brain regions (Fig. 5d-e), with one of the AstA neurons forming up to seven connections with one of six different FBl6 neurons. Since Lk and DH44 neuron tracing data is currently not available, we could not determine if Lk and DH44 neurons also form synaptic connections with FBl6 neurons. Published light-level expression patterns for Lk^{80, 81} and DH44^{11, 82} neurons show extensive projections from these neurons in the SMP, SLP, and SIP regions. Since FBl6 neurons also have extensive arborization in the SMP, SIP, and SLP brain regions (Fig. 5b-c), we predict that it is in these higher brain regions that Lk and DH44 neurons also form synaptic connections with FBl6 neurons. The signals that FBl6 receives from these upstream modulatory neurons are then integrated by FBl6 in a state- and context-dependent manner to generate a transient and dynamic representation of choice, which is relayed to downstream motor circuits (Fig. 5f).

Line 306: food preference was not affected by manipulations of TPN3: is there any olfactory component to the behavior?

While there could be an olfactory component to the behavior if olfactory cues were provided in the decision assay, we excluded olfactory cues from the decision context to control the number of variables addressed in this study. Since TPN3 is a secondary taste projection neuron, we tested whether it was involved in relaying relevant taste sensory information to FBl6 for behavior reported in this study.

Line 319: “Lack of FBL6 inhibition in response to a taste stimulus could represent acceptance of that option”. If FBL6 neurons are normally inhibited by tastants that they do not choose, how does inhibition of FBL6 with GPCR alter food preference towards bittersweet? (connected to comment 2 above—what is the conceptual model?)

Based on new behavior data (**Fig. 5a**, detailed above in relation to comment 2), we have now further discussed how changing the state of FBl6 neurons, by persistent optogenetic inhibition, during decision making could alter behavioral outcomes. These new results and our refined model are described as follows:

New text in Line 317-353:

FBl6 neurons are strongly inhibited by a single stimulation with rejected food choice (Fig. 4c-d), this inhibition is context-dependent, and persistent sensory-stimulus independent optogenetic inhibition of FBl6 neurons throughout the decision assay shifts food preference to bittersweet (Fig. 3a left panel, Supplementary Fig. 3b left panel). We thus hypothesized that imposing unnatural, temporally and spatially homogenous optogenetic inhibition of FBl6 throughout the decision task disrupts value integration and decision making. This imposed activity pattern, that does not take into account the sensory environment and current value estimates of the fly in any given moment, results in the hungry fly defaulting to higher calorie bittersweet food. We tested this model further by optogenetically inhibiting FBl6 neurons on only one side of the arena while presenting flies with the same sweet food on both sides. We found that control flies not expressing GtACR1 avoided the lit side of the arena indicating intrinsic avoidance of bright green light⁷⁵. In contrast, persistent FBl6 optogenetic inhibition mediated by GtACR1 abolished this bright green light avoidance, resulting in equal preference for food on both sides (Fig. 5a, Supplementary Table 1). This indicates that persistent optogenetic inhibition of FBl6 induces a positive preference that counteracts the intrinsic aversion to bright green light. While spatially and temporally homogeneous optogenetic inhibition of FBl6 neurons during decision making shifted preference to bittersweet (Fig. 3a left panel, Supplementary Fig. 3b left panel), spatially restricted optogenetic inhibition shifted preference to sweet (Fig. 5a). Congruent with context-dependent FBl6 neural response to taste stimuli (Fig. 4c-d, Supplementary Fig. 4a-d), optogenetically inhibiting the same FBl6 neurons in different behavioral contexts produced different choice outcomes.

These findings are inconsistent with predictions of a simplistic ON/OFF sensorimotor model in which FBl6 neurons would turn feeding on and off on command independent of context. Instead these findings support a model in which the activity of FBl6 neurons is transient, dynamic, and highly state- and context-dependent. We hypothesize that, similar to mammalian nucleus accumbens and other cortical circuits involved in gating the flow of information to cortex for attention and decision-making^{76, 77, 78}, FBl6 neurons are bistable and oscillate between two states that are characterized by different values of membrane potential. According to the synaptic gating model^{76, 77, 78}, in the “down state”, FBl6 neurons would be hyperpolarized (i.e., closed gate) leading to rejection of an option. To reach the “up state”, i.e., a membrane potential just below the neuron’s firing threshold, FBl6 neurons would have to receive input from another neuron. Only when FBl6 neurons are in this “up state” (i.e., open gate) can subsequent inputs trigger action potentials leading to acceptance of an option influenced by these inputs. Similar to mammalian cortical gatekeeper neurons^{76, 77, 78}, FBl6 neurons could act as a gatekeeper that integrates converging inputs from upstream neuromodulatory neurons using non-linear summation (Fig. 5f). If the activity of FBl6 neurons were forced into an unnatural “down state” using optogenetic inhibition, the integration of inputs to FBl6 neurons would be disrupted. Under such a condition, hardwired innate inputs could be shunted to bypass the gatekeeper circuit motif resulting in unexpected behavioral outcomes that would not take into account adaptive flexible input variables such as the context in which flies choose food options.

**Line 597: “3-5 day old flies (naïve or after two-choice assay)”:
How much time passed between the two-choice assay and calcium imaging?**

This is an important experimental parameter, as recognized by this reviewer, and between 20 min and 1.5 hr elapsed between the behavior assay and Ca²⁺ imaging. This time range was required for appropriately immobilizing flies in the recording chamber without anesthesia, followed by cuticle dissection and recovery after dissection. We have now added this information to the Methods section.

What is the difference between “fed flies” and “two-choice assay” flies considering both groups recently ate food?

We have now explicitly provided this information in the Methods section,

New text in Line 524-527:

Fed flies ate regular cornmeal food, were never food deprived, and didn't experience the two-choice assay. Flies that experienced the two-choice assay were food-deprived for 21h and chose to eat agar based sweet or bittersweet food (not regular cornmeal food) for only 5 min during the assay.

Fig2, panel b: how was lack of feeding quantified?

For each trial, flies that did not consume either food had no food dye in their abdomen, and were counted as having not fed. This information has been added in Methods section.

New text in Line 501-502:

We also recorded the number of flies that did not consume either food in the assay by counting the number of flies that had neither food dye in their abdomen.

Fig 4, panel c: how does this compare to c205-Gal4?

Published reports of light-level fluorescence microscopy data show complete overlap of c205-GAL4, 84C10-GAL4, and 23E10-GAL4 lines in FB16 (Donlea et al. Neuron, 2014, 2018, Nguyen, Rosbash lab thesis 2017). In addition, the expression pattern of 23E10-GAL4 and 84C10-GAL4 lines almost completely overlap in FB16 according to hemibrain EM connectome data. We have now performed new experiments confirming our receptor RNAi results using 84C10-GAL4 (**Suppl. Fig. 3b** right panel). In an important new control experiment, we have now also shown that deletion of FB16 expression from the c205 expression pattern using 84C10-LexA and LexAop-GAL80 abolishes the shift in food preference induced by optogenetic inhibition using c205 on its own (**Fig. 3a**, “84C10>GAL80+c205>Gt”).

Fig. 4, panel d: when do these responses return to baseline?

We stopped recording 3-4 s after the removal of taste stimulus to reduce photobleaching of GCaMP. We removed traces of food left on the fly foreleg between each trial before starting the next trial. Responses likely returned to baseline during the 15 s inter-trial interval, as we were able to record robust inhibitory responses during the next trial.

Fig. 4, panel d-e: why are the bittersweet responses in ‘chose bittersweet’ so variable compared to all other groups? Should increase the sample size slightly.

As suggested, we have increased the sample size and removed the one outlier that was causing large variance in the ‘chose bittersweet’ dataset (due to a technical artifact in the measurement represented by this outlier). Note that the error bars depict 95% CI, which are larger than SEMs, but provide more information on the underlying distribution.

Reviewer #2 (Remarks to the Author):

The manuscript by Sareen et al., describes a set of very interesting and insightful experiments that help identify a novel region of the brain in decision making. It is well written for the most part. The authors begin with devising a robust choice-test that measures decision making in hungry flies. With this test they screened a series of neuronal classes that are likely to be implicated in decision making in the context of the taste stimuli and the internal state of the fly. They identify a limited and interesting set of neuropeptides (NPs; many of which have been implicated in feeding behaviour earlier), three sets of dopaminergic neurons and very interestingly a higher brain centre – the fan shaped body (FB) from the central complex. Using optogenetics and a knockdown approach for neuropeptide and dopaminergic receptors they identify putative pathways for information flow amongst the various neuronal classes. Similarly they place some NPs as directly regulating activity of FB neurons whereas others as sending information indirectly to the FB neurons. Finally, by directly measuring activity in the FB neurons during a choice test they show that increased activity correlates with the behavioural choice to a sensory stimulus. While the data are convincing, given the complexity of the subject there are several aspects that need addressing and clarification before publication:

We thank Reviewer #2 for their insightful comments. We have performed suggested experiments, added requested discussion, and cited all of the suggested literature. Please see our point-by-point response below.

1) Where do the authors place the Leucokinin neurons that have earlier (Yurgel et al 2019 Plos Biol) been shown to modulate feeding state?

Neuropeptides perform various functions that are highly context and state-dependent. For example, the same authors as Yurgel et al., PLoS Biol 2019 show in their earlier study (Zandawala, Yurgel et al., PLoS Genetics 2018) that Lkr mutants display opposite behavior from Lk mutants, that is, increased PER instead of decreased PER in response to sugar in hungry flies. Zandawala, Yurgel et al. hypothesize that a different receptor or downstream signaling pathway may be responsible for the opposite phenotype. Therefore, we place our results in the specific context of the behavior that we assay in this study, i.e., under sweet-bittersweet gustatory conflict. Based on our findings, we hypothesize that Lk neurons encode and relay metabolic state information to FB16, Line 400-402:

Lk neurons have been implicated in nutrient sensing^{10, 81}, and their activity regulates feeding in food deprived flies (Fig. 2a-b), suggesting that internal metabolic state information could reach FBL6 through Lk/Lkr signaling.

We have now also cited the study suggested by the reviewer in our Discussion.

2) Allatostatin neurons are thought to inhibit feeding in adult *Drosophila* (Hergraden et al., PNAS 2012). However, activation of AstA neurons by Chr does not appear to change food intake in the assay shown here. This needs to be addressed in the context of how AstA helps resolve sensory conflict during feeding.

This is related to our explanation for comment 1. Neuropeptides perform various functions that are highly context and state-dependent. Accordingly, there are several differences in Hergarden et al. and our assay that present different contexts to flies. Under these different contexts, it is not surprising that AstA neurons may have different roles. For example, Hergarden et al., used CAFÉ. The surface area covered in food and therefore available to all flies at all times was much smaller in Hergarden et al. compared to our assay, in which all flies always sensed food because they were standing on it. A lower probability of interaction with food could explain the lower percentage of flies consuming food in Hergarden et al. In addition, Hergarden et al. chronically activated AstA neurons with NaChBac as opposed to the temporally restricted acute activation with CsChrimson in our study. In addition, there are other studies showing that flies do consume food during AstA activation (Hentze et al., Scientific Reports 2015). Our finding that AstA activation decreases preference for bittersweet food (**Fig. 2c**) is actually congruous with Hergarden et al. and Hentze et al. in that AstA activation seems to increase satiety. In our assay, this increased satiety corresponds with the rejection of higher-calorie bittersweet food option, and consumption of lower-calorie sweet food.

Taken together, we expect that the activity of all the neuromodulators tested in our study, including AstA, affect behavior in a context- and state-dependent manner having different effects in different behavioral contexts. Our results are obtained in the context of binary choice between sweet and bittersweet food options, and we consider them neither in contradiction nor support of what neuropeptidergic neurons do in other contexts.

As suggested, we have added more discussion of how neuropeptides can have highly context and state-dependent roles and have cited the suggested paper.

New text in Line 134-145:

Neuropeptides perform diverse functions that are highly state- and context-dependent. For example, Lk and Lkr mutants have been shown to have opposite effects on proboscis extension response in response to sugar in hungry flies⁺, which have been hypothetically attributed to different receptor or downstream signaling pathways. Mammalian NPY producing AgRP neurons are also known to have highly state- and context-dependent roles in feeding behaviors. Recent studies have challenged the textbook model and shown that AgRP neurons are inhibited within seconds by sensory detection of food even though on the longer timescale AgRP neurons are activated by energy deficit and promote food consumption^{58, 59, 60}. Similarly, mammalian hypothalamic POMC neurons were initially

thought to encode satiety, but recent studies have found that their activation can also promote feeding⁶¹. The effects that different neuropeptidergic neurons have in the food-related choice assay employed here are thus concluded to be specific to the context of sweet-bittersweet gustatory decision making.

3) The change in preference by activating and inhibiting AstA neurons is remarkable. Did they try expressing AstA RNAi with the activity inhibiting transgene (Gt)? Do they expect the shift towards bitter-sweet to be lost as well?

We have not expressed AstA-RNAi in AstA neurons while also inhibiting AstA neurons with GtACR1. However, both manipulations individually have the same effect of reducing secretion of AstA. Therefore, we expect the combined effect to be the same as inhibition of AstA neurons alone, resulting in shift towards bittersweet preference.

And would overexpression of AstAR on FB neurons shift food preference?

This is an interesting question. Since AstA receptor knockdown in FB16 neurons shifts preference to bittersweet (**Fig. 3a**), in line with the reviewer's hypothesis, overexpression of AstA receptors could shift the preference towards sweet. This would be consistent with our finding that optogenetic activation of AstA neurons shifts the preference to sweet (**Fig. 2c**). Although we have not conducted these experiments, the outcome of such manipulation will also depend on factors that impose rate-limiting and ceiling effects on the response of a neuron. Such factors would include the number of receptors coupled to downstream signaling pathways that are required to achieve a specific signal in a state- and context-dependent manner, and whether AstA peptide levels, rather than AstAR receptor levels, are limiting for downstream intracellular signaling pathway activation.

4) Several of the neuropeptide GAL4s change behaviour by either activation or inhibition but NOT both. Is it possible that these NPs help recognise the valence of a taste stimulus i.e DH44 helps recognise bitter taste? This needs to be explained better for all the identified NPs, where the shift in behaviour is with either activation or inhibition but NOT both (Figure 2).

One likely explanation for behavior effects induced by either optogenetic activation only or inhibition only of NPF and DH44, but not both, is if a neuron is already very active or inactive in the particular context, then optogenetic activation or inhibition, respectively, is expected to have no effect. Only in contexts where a neuron is in a "mid-range" of activity is either activation or inhibition expected to have an effect. Understanding how and why these bidirectional effects only occur with some neuropeptidergic neurons and not others would require further investigation that is beyond the scope of the current study. However, we do have specific hypotheses about what these different neuropeptidergic neurons may encode. Based on published roles of these neuropeptides and their effects on food preference in our study, we hypothesize that NPF neurons help recognize taste/food stimulus valence such as bitter taste, while DH44 and AstA neurons help recognize food content or identity such as protein vs. carbohydrate. These hypotheses and their explanations are described in our Discussion.

New text in Line 390-402:

Upstream neuromodulatory subsets in the decision ensemble we identified have known roles in hunger dependent food intake^{7, 10, 12, 83}, reward⁸⁴, valence^{22, 62}, and long-term memory^{18, 22, 62}. Since under natural foraging conditions, flies conduct food search while walking and flying, it is expected that sensory information about food seeking during flight would also be integrated in this ensemble. Consistently, we also identified flight-promoting dopaminergic neurons involved in food seeking behaviors^{31, 63}. These modulatory neurons are well positioned to estimate value of features in the sensory environment and internal state that can be updated in a state- and context-dependent manner. For example, AstA neuron activity influences relative carbohydrate and protein preference⁷, while DH44 neurons sense sugars⁸³ and amino acids¹². Thus, AstA and DH44 neurons could convey food identity information to FBl6. NPF neuron activation is inherently rewarding⁸⁴, and thus could convey food valence information. Lk neurons have been implicated in nutrient sensing^{10, 81}, and their activity regulates feeding in food deprived flies (Fig. 2a-b), suggesting that internal metabolic state information could reach FBl6 through Lk/Lkr signaling.

5) The activation of DA subsets shifted preference towards bitter-sweet but activation/inhibition of the MB regions to which these neurons project had no effect on the choice – this contradictory result needs to be explained. At least some of these DA subsets project to other regions of the brain – e.g The PPL1-g2a'1 DANs project to the PAM neurons (Felsenberg et al., 2017), as well as the fan shaped body (FSB) and the lateral accessory lobe (LAL; Scaplen et al., 2020). Could these alternate projections be relevant in the context of choice?

As noted by the reviewer, DA neurons identified in our study project to several other brain regions in addition to the MB. We predict that it is some of these alternate dopaminergic projections that are relevant for decision making in the context of our study. Since the same DA neurons have roles in a variety of behaviors, not all synaptic connections they have can be relevant to all of these behaviors. E.G., even though there are direct synaptic connections between dopaminergic neurons and FBl6 that are relevant for sleep and ethanol preference, DA receptor RNAi in FBl6 had no effect on sweet-bittersweet food choice. Rather, DA receptor RNAi in the neuropeptidergic neurons influenced sweet-bittersweet food choice. We thus hypothesize that DA neurons influence sweet-bittersweet food choice by modulating the neuropeptidergic neurons via dopaminergic terminals in the SMP, SLP, and SIP brain regions, which are also innervated by neuropeptidergic neurons. As, suggested, we have elaborated our discussion of how DA neurons could modulate the decision making ensemble.

New text in Line 151-156:

The dopaminergic neurons influencing sweet-bittersweet food choice have projections in several other brain regions in addition to mushroom body, such as, superior medial protocerebrum (SMP), superior lateral protocerebrum (SLP), and superior intermediate protocerebrum (SIP). We thus conclude that dopaminergic projections to these other non-mushroom body brain regions mediate dopaminergic modulation of sweet-bittersweet decision making.

New text in Line 223-228:

While FBl6 neurons receive synaptic input from dopaminergic neurons^{46, 47} that regulate sleep^{46, 47} and ethanol preference⁴⁹, interestingly, direct dopaminergic input to FBl6 through dopamine receptors did not influence food choice (Fig. 3a right panel). Instead, we hypothesize that the dopaminergic neurons we identified regulating food choice modulate the activity of AstA, DH44, and NPF neurons (Fig. 2b-e), and thereby indirectly influence FBl6 neurons to modulate sweet-bittersweet choice.

New text in Line 409-416:

Dopaminergic subsets involved in aversive memory and modulation of flight and feeding (PPL1 $\gamma 2\alpha'1$)^{62, 63}, taste conditioning (PPL1 $\alpha 3$)²², and long-term memory (PAM $\alpha 1$)¹⁸ also affect food choice in the decision assay (Fig. 2a), and could provide signals for predicting and updating value estimates in working memory, analogously to primate dopaminergic ventral tegmental area⁸⁵. Since the identified dopaminergic neurons project to SMP, SLP, and SIP regions, where the neuropeptidergic neurons also arborize and connect to FBl6 neurons, it is conceivable that it is in these same higher brain regions that the identified dopaminergic neurons directly modulate neuropeptidergic neurons (Fig. 2b-e) to update value estimates encoded by them.

6) Activation of TRH GAL4 leads to no feeding. Is this expected? The implication of this result needs explaining.

Pooryasin and Fiala, J Neuro 2015, have reported that activating large populations of serotonergic neurons, using Trh-GAL4, induces quiescence and inhibits feeding and mating in flies. They also identified specific neural clusters that suppress mating, but not feeding, suggesting that serotonin does not uniformly act as a global, negative modulator of general arousal. Our optogenetic activation experiment with Trh-GAL4 reproduces these previously published results, in that flies ceased movement (i.e., were quiescent) on optogenetic activation of Trh-GAL4, which resulted in no feeding (**Suppl. Fig. 2a**). This is now explained in the Results.

New text in Line 156-161:

Additionally, optogenetic activation of a large population of serotonergic neurons targeted by Trh-GAL4 ceased fly movement and resulted in no feeding (Supplementary Fig. 2a). A previous study has also reported that activating large populations of serotonergic neurons using Trh-GAL4 induces quiescence and inhibits feeding and mating in flies. This study also identified specific neural clusters that suppress mating, but not feeding, suggesting that serotonin does not uniformly act as a global, negative modulator of general arousal⁶⁴.

7) In naïve hungry flies bitter-sweet ingestion correlates with inhibition of FB neurons whereas sweet had no effect. Given that hungry flies eat both types of food equally when hungry, this result needs better explanation.

We agree that our language did not provide an intuitive interpretation of these results. We would like to clarify that **Fig. 4c-d** depict FB16 neural responses to only one stimulation with sweet and bittersweet taste in naïve flies, which is different from the conflicting choice assay context. In the choice assay, flies are free to sample both food options repeatedly before consumption. With new animal tracking analyses using recorded videos of flies performing the assay, we have now shown that almost all flies in our choice assay visit both food sectors throughout the assay and transition from one to the other multiple times (**Suppl. Fig. 1g-j**). Even though almost all individual flies sample both options, they consume only one type of food. In the equal-preference condition roughly half of the flies eat only sweet and half eat only bittersweet food. In other words, it is not that individual hungry flies eat both foods equally, but that half of hungry flies eat only sweet and half eat only bittersweet at the equal preference condition.

Since individual flies eat only one type of food in the assay, we could map FB16 neural responses to sweet and bittersweet taste to each fly's individual choice in the preceding behavioral assay. Our results show that inhibitory FB16 responses correspond with rejected choice in all other conditions that we tested (**Fig. 4c**), and flies innately find bittersweet food aversive (Masek, Scott, PNAS 2010, French et al. J Neuro 2015, Lee, Moon and Montell, PNAS 2009, Sellier et al., Chem Senses 2011, French et al., J. Neuroscience 2015). We thus posit that strong FB16 inhibition by one stimulation with bittersweet taste in naïve food-deprived flies represents rejection of this stimulus. We have now added this expanded explanation to the Results section (Line 270-273).

We have also added a description of how the preference index was calculated at the beginning of the Results section to aid interpretation of results (new text in Line 52-55), and changed language to clarify what equal-preference means (new text in Line 63-70).

New text in Line 52-55:

Food choice was quantified by scoring the color of the food in each fly's abdomen at the end of the assay. A preference index was then calculated by subtracting the number of flies that ate bittersweet food from the number of flies that ate sweet food divided by the total number of flies that consumed food in that trial.

New text in Line 63-70:

At 10- and 5-fold sucrose concentration ratios (Fig. 1b, 50 mM vs. 500 mM sucrose + 1 mM quinine, 100 mM sucrose vs. 500 mM + 1 mM quinine at 21 h) almost all flies consumed only a single food option with half of them consuming only sweet and half consuming only bittersweet resulting in equal-preference for sweet and bittersweet at these conditions. The equal-preference point varied with food-deprivation duration, shifting preference from sweet toward bittersweet with increasing food-deprivation duration (Fig. 1b, Supplementary Fig. 1d-f, Supplementary Table 1), suggesting an external taste sensation and internal hunger state equilibrium at the equal-preference concentration ratios.

New text in Line 270-273:

Flies innately find bittersweet food aversive²⁷, and reduce proboscis extension response to and consumption of bittersweet mixtures^{66, 67, 68}. Thus inhibitory taste responses in FBl6 to the first encounter with bittersweet food may represent rejection of this option.

8) Fig 4f leaves out the various classes of DA neurons identified in Fig 2 – are they equivalent in their connection to the NP neurons? This is not clear and should be addressed.

We have now updated our **Fig. 5f** schematic to include three specific classes of DA neurons that we have identified to be important for food preference in our study. We expect the connections between DA neurons and neuropeptidergic neurons to be involved in dynamic updating of value estimates of food and hunger signals encoded by the neuropeptidergic neurons. However, we have not yet mapped out which DA neurons are functionally connected to which neuropeptidergic neurons in our behavioral context. Unraveling connectivity of various DA neurons to neuropeptidergic neurons and function of these connections is very interesting. Investigating these connections would entail activating and inhibiting each DA neuron separately while measuring neural activity in each neuropeptidergic neuron. Behavioral effects of manipulating each identified functional connection would then need to be tested. While very interesting, such detailed investigation is beyond the scope of the present study. We have added more discussion of how and where DA neurons modulate neuropeptidergic neurons at lines 365-352.

New text in Line 409-416:

Dopaminergic subsets involved in aversive memory and modulation of flight and feeding (PPL1 $\gamma 2\alpha '1$)^{62, 63}, taste conditioning (PPL1 $\alpha 3$)²², and long-term memory (PAM $\alpha 1$)¹⁸ also affect food choice in the decision assay (Fig. 2a), and could provide signals for predicting and updating value estimates in working memory, analogously to primate dopaminergic ventral tegmental area⁸⁵. Since the identified dopaminergic neurons project to SMP, SLP, and SIP regions, where the neuropeptidergic neurons also arborize and connect to FBl6 neurons, it is conceivable that it is in these same higher brain regions that the identified dopaminergic neurons directly modulate neuropeptidergic neurons (Fig. 2b-e) to update value estimates encoded by them.

9) The changes in FB neuron activity and their correlation with a choice suggests that FB neurons might have high basal activity. Is this the case and is their basal activity different in a hungry and fed fly? It would be helpful if basal activity measures of FB neurons are shown and mentioned clearly.

As suggested, we have performed new experiments to compare basal Ca^{2+} levels of FBl6 neurons between naïve hungry and fed flies and find no differences. These data are shown in **Suppl. Fig. 4e-h**, with non-normalized individual fly traces shown in **Suppl. Fig. 4g-h** and described in Line 276-278:

Basal FBl6 neural Ca^{2+} levels in the absence of taste stimulation was indistinguishable between naïve hungry and fed flies (Supplementary Fig. 4e-h).

Although it is not straightforward to measure basal neural activity using calcium imaging, as pointed out by the reviewer, strong sensory-evoked decreases in intracellular Ca^{2+} do indeed suggest high basal neural activity.

10) When PPL1- $\gamma 2\alpha'1$ DANs are inhibited, starved flies take much longer to identify a food source (Tsao et al., eLife, 2018; Sharma and Hasan, eLife, 2020). Do the authors think that the decision not to search for food in a hungry fly (in the absence of a choice) also requires activity in FB neurons? Or is the role of FB neurons only when there is a choice of food? This aspect should be included in lines 325-326 of the discussion.

This is an interesting point. FBL6 neurons are involved in accepting and rejecting food options, integrate hunger state into this decision, and influence food-related behavior in a no-choice context (**Fig. 5a, Suppl. Fig. 5**). It is thus possible that these neurons could also be involved in making a decision to not search for food in the absence of a choice when flies are sated. We have now added discussion and cited the suggested references in our manuscript.

Citation in Line 92-95:

Various neuromodulatory and other neurons regulate hunger dependent food intake^{7, 8, 9, 10, 11, 12, 13}, hunger dependent odor encoding and food search^{14, 15, 16}, reward^{17, 18, 19, 20} or punishment²¹, memory^{18, 20, 22}, and internal state for courtship^{23, 24, 25}.

Next text and citation in Line 358-363:

Hungry flies take longer to identify a food source when specific dopaminergic neurons (PPL1 $\gamma 2\alpha'1$) are inhibited^{31, 63}. Since FBL6 neurons are involved in accepting and rejecting food options, integrate hunger state to inform this decision, and affect food-related behavior in a no-choice context (Fig. 5a, Supplementary Fig. 5), it is conceivable that these neurons could be involved in making a decision to not search for food in the absence of a choice when flies are sated.

11) Finally, the authors mention how integration of upstream inputs at the FB very likely determines foraging in natural conditions. Under natural conditions flies forage by walking and by flying. Thus, any neurons that integrate information for making a foraging decision would also need to integrate sensory information from flight promoting circuits (Sharma and Hasan eLife, 2020) and this aspect should be included in the discussion.

Gaiti Hasan

As suggested, we have now included this aspect in our discussion and cited the suggested reference.

New text in Line 392-395:

Since under natural foraging conditions, flies conduct food search while walking and flying, it is expected that sensory information about food seeking during flight would also

be integrated in this ensemble. Consistently, we also identified flight-promoting dopaminergic neurons involved in food seeking behaviors^{31, 63}.

Reviewer #3 (Remarks to the Author):

In this study, Sareen et al. suggest that they have identified a subset of fan-shaped body (FB) neurons in the fly brain that integrates taste quality, previous experience, and hunger state to encode behavioral decisions. They devised a food choice assay, allowing flies to choose between sweet-only and bittersweet food. They show that flies exhibited no preference between sweet-only and bittersweet food if the sucrose concentration in the bittersweet food was ten times higher than that in the sweet-only food. Under these equal-preference conditions, the authors searched for neurons whose activation or inhibition shifted the preference towards sweet-only or bittersweet food based on the idea that such neurons are likely involved in the decision-making process during flies' food choices. They tested various neurons previously shown to play a role in learning, memory, decision-making, navigation, and hunger-driven behavior. Their positive hits represent several types of feeding- or hunger-related peptidergic neurons (including those expressing LK, AstA, NPF, or DH44), three mushroom body-innervating dopaminergic neurons (PPL1- γ 2 α '1, PPL1- α 3, and PAM- α 1), and a small subset of FB layer 6 (FBI6) neurons. Using RNAi against the receptors for the neuropeptides and dopamine, the authors mapped the potential connectivity between these neurons. Their data suggest that FBI6 neurons receives direct inputs of AstA, DH44, and Lkr and thereby acts as an integration node for these three neuropeptide signals. Finally, the authors have presented a curious scenario whereby the Ca²⁺ response in FBI6 neurons to food taste can predict the food choice made earlier by the flies. Presenting food that had previously been rejected by the fly to its foreleg caused a decrease in Ca²⁺ signal in those FBI6 neurons. Therefore, the authors conclude that FBI6 encodes "rejected choice."

Overall, the experiments presented in this manuscript have been done methodically and the presented data is of high quality. The phenomenon they have observed in this study is interesting and warrants further investigation. However, I have serious concerns about the authors' data interpretation and am not convinced that their results support the conclusions they make in this manuscript.

We thank Reviewer #3 for their comments and insightful suggestions for improvement. We have performed all suggested experiments, added requested discussion, and cited all of the suggested literature. Please see our point-by-point response below.

My first concern is their food choice assay in which flies chose bittersweet food (500 mM sucrose + 1 mM quinine) over sweet-only food containing < 50 mM sucrose. However, when the sucrose concentration in the sweet-only food was > 50 mM, more flies chose it. The authors have interpreted this finding as indicating that "the caloric advantage in choosing a less palatable bittersweet food is outweighed by the danger-avoidance advantage of the sweet option." However, no evidence is presented that the bittersweet food is less palatable than the sweet-only food when the sucrose concentration in the sweet-only food is low.

Flies innately find bitter compounds (including quinine) mixed with sugars aversive (Lee, Moon and Montell, PNAS 2009, Sellier et al., Chem Senses 2011, French et al. J Neuro 2015) and less palatable without the presence of a sweet only choice, which is demonstrated by reduced proboscis extension response to and consumption of bitter-sweet mixtures (Masek, Scott, PNAS 2010, French et al. J Neuro 2015, Lee, Moon and Montell, PNAS 2009, Sellier et al., Chem Senses 2011, French et al., J. Neuroscience 2015). This well-established effect of bitter adulteration on palatability guides our interpretation of our results.

Isn't it equally possible that the 500 mM sucrose masks the bitterness of quinine at the sensory level so that the bittersweet food is more or equally palatable to the sweet-only food of low sucrose concentration?

While sensory masking is generally possible, hunger level, which changes with food-deprivation duration, shifts food preference in our assay (**Fig 1b**, compare 100 mM between 2h, 6h, and 21h deprivation durations). We have now provided statistical tests, figures, and data demonstrating that hunger shifts equal-preference in our assay (**Suppl. Data Fig. 1d-f**, also see a detailed response below to the next related comment). Since these results demonstrate that hunger shifts the equal-preference condition to different sucrose and quinine concentration ratios, this equal-preference cannot be solely explained by hunger-independent sensory masking of bitter quinine with 500 mM sucrose.

Furthermore, the authors suggest that the hungriiness of the fly regulates this food choice behavior and that hungrier flies exhibit a stronger preference for the bittersweet food (L136-137). However, their results in Fig. 1b do not support this claim, showing no shift in food preference when flies had been starved for longer. The authors use this assay to probe a complex decision-making process involving value tradeoff, internal states, sensory inputs, and past experiences, but they need to provide more evidence supporting its appropriateness.

We agree that our explanation and analyses of hunger effects on food choice were not stated as clearly as possible. We have now clarified this with edits to the text, new data, and new statistical analyses.

1. Our data in **Fig. 1b** suggest that equal preference for sweet and bittersweet is extended from the 50 mM condition to the 100 mM condition when flies are food deprived for 21 h (**Fig. 1b**, **Suppl. Fig. 1d**). To determine if this apparent shift in preference from sweet to equal preference significantly varied with food deprivation duration, i.e., preference shifted from sweet towards bittersweet with hunger level, we performed one-way ANOVA followed by post-hoc test for linear effect in the data. We found that the slope of preference index over deprivation time was negative and this slope was statistically different from zero (**Suppl. Fig. 1d**). Statistically significant linear effect in these data with a negative slope indicates decreasing preference for sweet as flies are food deprived from 2 h to 21 h (**Suppl. Fig. 1d**, slope = -0.165, p = 0.0149).
2. We have also provided new additional data in **Supplementary Fig. 1e-f** from wild-type strain Canton S, (CS). Wild-type CS flies prefer sweet at the 50 mM sucrose vs. 500 mM sucrose + 1 mM quinine condition when food deprived for only 5h compared to equal

preference for sweet and bittersweet when food deprived for 21h. This difference in preference is statistically significant (**Suppl. Fig. 1f**, unpaired t-test $p = 0.016$). These new data, in a wild-type strain distinct from the *w1118* control strain required for genetic experiments, establish that the equal-preference point does change with hunger state. Note that ANOVA with post-hoc linear effect analysis or t-test are employed in these two contexts as appropriate for the number of durations of food deprivation being compared.

Together, these data demonstrate that equilibrium point for food preference shifts with hunger state.

The authors show that silencing of DH44- or Lk-expressing neurons shifted flies' food preference towards bittersweet food, whereas activation of NPF neurons shifted the preference towards sweet-only food. Even if we tentatively accept that preferring bittersweet food indicates a higher hunger level, these results do not agree with several previous studies. NPF is the fly homolog of mammalian neuropeptide Y (NPY). Both NPF and NPY have been shown to be hunger signals that promote feeding and food-seeking behavior (Wu et al., Nat. Neurosci. 2005 & PNAS 2005; Inagaki et al., Neuron 2014; Bhagyashree et al., Nat. Neurosci. 2019; Beshel & Zhong, J. Neurosci. 2013; Krashes et al., Cell 2009; Tsao et al., eLife 2018). Therefore, activation of NPF neurons should shift flies' food preference to the bittersweet rather than sweet-only option.

We agree that our NPF findings could be interpreted as contradictory to previously published findings. However, neuropeptides are involved in various computations and behaviors that are highly context- and state-dependent. It would be hard to explain how the brain performs so many different computations with a limited set of neuromodulators if one neuropeptide or neurotransmitter performed only a singular function. For example, recent studies have challenged the textbook model of the role of mammalian NPY producing AgRP neurons in feeding behaviors. AgRP neurons are inhibited within seconds by sensory detection of food (Chen et al., Cell 2015, Betley et al., Nature 2015, Chen et al., eLife 2016) even though on the longer timescale AgRP neurons are activated by energy deficit and promote food consumption (see also response to related comments for DH44 and Lk below), demonstrating that these neurons have highly context- and state-dependent roles. Likewise, mammalian hypothalamic POMC neurons were initially thought to encode satiety, but recent studies have found that their activation can also promote feeding (Koch et al., Nature 2015). We thus interpret our results in the specific context of sweet-bittersweet decision making, and interpret them as neither in contradiction nor support of what NPF neurons do in other contexts. We have added more discussion to elaborate on this point.

New text in Line 134-145:

Neuropeptides perform diverse functions that are highly state- and context-dependent. For example, Lk and Lkr mutants have been shown to have opposite effects on proboscis extension response in response to sugar in hungry flies+, which have been hypothetically attributed to different receptor or downstream signaling pathways. Mammalian NPY producing AgRP neurons are also known to have highly state- and context-dependent roles in feeding behaviors. Recent studies have challenged the textbook model and shown that AgRP neurons are inhibited within seconds by sensory detection of food even though

on the longer timescale AgRP neurons are activated by energy deficit and promote food consumption^{58, 59, 60}. Similarly, mammalian hypothalamic POMC neurons were initially thought to encode satiety, but recent studies have found that their activation can also promote feeding⁶¹. The effects that different neuropeptidergic neurons have in the food-related choice assay employed here are thus concluded to be specific to the context of sweet-bittersweet gustatory decision making.

Similarly, DH44 neurons are activated by nutritious sugar to promote food intake (Dus et al., Neuron 2015). Accordingly, silencing DH44 neurons is expected to make flies favor sweet-only food.

While DH44 neurons have been shown to sense sugars and amino acids by increasing neural activity in response to these compounds (Dus et al., Neuron 2015), what happens when DH44 neurons sense the presence or absence of sugar during sweet-bittersweet decision making has not been previously addressed. Additionally, one could argue that since DH44 neurons are activated by nutritive sugars in the hemolymph (Dus et al., Neuron 2015), absence of such detection of sugar in a hungry fly could inhibit DH44 neurons. In such a scenario, inhibition of DH44 neurons would represent hunger and would thus be predicted to increase intake of higher calorie food, which would be consistent with our findings.

Our findings demonstrate the role of DH44 neurons in the context of sweet-bittersweet food choice and internal state of the fly tested here. In our model for decision making under sensory conflict, we hypothesize that DH44 neurons encode and convey food identity information to FB16 neurons (**Fig. 5f**), and are not an ON/OFF switch for food intake.

Finally, two recent studies (Yurgel et al., PLoS Bio. 2019; Bhagyashree et al., Nat. Neurosci. 2019) have shown that some LK neurons are activated by starvation to drive feeding and food-seeking behavior. Activation of LK neurons also decreases postprandial sleep (Murphy et al., eLife 2016). These studies suggest that LK is a hunger signal, which is inconsistent with the findings of the current manuscript implying increased perceived hunger upon inhibition of LK neurons. Consequently, I feel it is important for the authors to compare and discuss their findings with those previous studies.

We would like to again emphasize that neuropeptides perform diverse functions that are highly context- and state-dependent (see also response to comment on NPF above). Authors cited by the reviewer reported in an earlier study (Zandawala, Yurgel et al., PLoS Genetics 2018) that Lkr mutants display opposite behavior from Lk mutants, that is, increased PER instead of decreased. They hypothesize that a different receptor or downstream signaling pathway may be responsible for the opposite phenotype. In addition, Bhagyashree et al., Nat Neurosci 2019, show that Lk neurons both activate and inhibit different dopaminergic neurons in a state-dependent manner to control sugar and water memory. Thus we interpret our results in the specific context of our sweet-bittersweet food choice.

As suggested, we have added more discussion of how neuropeptides can have highly context- and state-dependent roles and have cited the suggested papers.

New text in Line 134-145:

Neuropeptides perform diverse functions that are highly state- and context-dependent. For example, Lk and Lkr mutants have been shown to have opposite effects on proboscis extension response in response to sugar in hungry flies⁺, which have been hypothetically attributed to different receptor or downstream signaling pathways. Mammalian NPY producing AgRP neurons are also known to have highly state- and context-dependent roles in feeding behaviors. Recent studies have challenged the textbook model and shown that AgRP neurons are inhibited within seconds by sensory detection of food even though on the longer timescale AgRP neurons are activated by energy deficit and promote food consumption^{58, 59, 60}. Similarly, mammalian hypothalamic POMC neurons were initially thought to encode satiety, but recent studies have found that their activation can also promote feeding⁶¹. The effects that different neuropeptidergic neurons have in the food-related choice assay employed here are thus concluded to be specific to the context of sweet-bittersweet gustatory decision making.

The authors have studied the role of FBI6 neurons by mainly using c205-GAL4 in their behavioral experiments. The authors do not show its expression pattern, but c205-GAL4 also strongly labels a group of neurons in the subesophageal zone (SEZ), which is a convergence site for gustatory inputs (Hu et al., Cell Reports 2018). The authors should validate all their c205a-GAL4 results using 84C10-GAL4. They do present some behavioral experiments using 84C10-GAL4 in Extended data fig. 3, but they are not mentioned in the main text.

This raises an important point, since as noted by the reviewer, SEZ is the convergence site for inputs from gustatory sensory neurons. As noted, we had validated optogenetic activation and inhibition effects on food preference with 84C10-GAL4 (**Suppl. Fig. 3b** left panel), and we now highlight this in the main text. As suggested, we have now also added new data validating receptor RNAi results for the three relevant neuropeptides using 84C10-GAL4 (**Suppl. Fig. 3b** right panel). Additionally, in an important new control experiment, we have now shown that deletion of FBI6 expression from the c205 expression pattern using 84C10-LexA and LexAop-GAL80 abolishes the shift in food preference induced by optogenetic inhibition using c205 on its own (**Fig. 3a**, “84C10>GAL80+c205>Gt”). Since 84C10-GAL4 has very strong and specific expression in FBI6 only, with no SEZ expression, these results confirm that it is indeed the FBI6 neurons responsible for the effects that we report in this study. As suggested, we have now emphasized this important control and cited 84C10-GAL4 data from **Suppl. Fig. 3** in the main text.

The baseline preference in Extended data fig. 3b appears to be different from those shown in the main figures.

Yes, while it is true that baseline preference in the 84C10-GAL4 line appears different, we have used appropriate genetic background controls that control specifically for any differences in baseline behavioral phenotype. Importantly, the effects of optogenetic activation and inhibition on food preference are the same using 84C10-GAL4 as with c205-GAL4, when compared to appropriate genetic controls.

Moreover, there is no control group for the data in Extended data fig. 3d. The authors should consider performing these experiments again using more consistent conditions and appropriate controls.

We have now added the appropriate control data and the results are consistent with c205-GAL4 results. Specifically, there is no statistically significant difference in preference for lit versus dark quadrants in the absence of food between control flies and flies expressing CsChrimson or GtACR1 in FBI6 neurons.

There is a strong possibility that neurons in the SEZ are regulated by hunger signals to control feeding behavior. Therefore, it is also critical that the authors verify their receptor knockdown in the FBI6 experiments using 84C10-GAL4.

As suggested, we have now performed new experiments validating receptor RNAi results for the three relevant neuropeptides using 84C10-GAL4 (**Suppl. Fig. 3b** right panel). In an important key new control experiment, we have now also shown that deletion of FBI6 expression from the c205 expression pattern using 84C10-LexA and LexAop-GAL80 abolishes the shift in food preference induced by optogenetic inhibition using c205 on its own (**Fig. 3a**, “84C10>GAL80+c205>Gt”). Since 84C10-GAL4 has very strong and specific expression in FBI6 only, with no SEZ expression, these results further confirm that it is indeed the FBI6 neurons responsible for the effects that we report.

Finally, I have many questions regarding their calcium imaging experiments. First, it is not clear how the experiments were performed. Were flies immediately imaged after the choice assay or was there a waiting period? Were the flies that underwent the choice assay food-deprived? If yes, for how long? I feel the authors need to describe their experiments in more detail.

Flies were imaged between 20 min and 1.5 hr following the choice assay. This time range was required for appropriately placing flies in recording chamber without anesthesia, followed by cuticle dissection and recovery after dissection. All flies that underwent the sweet-bittersweet choice assay were food-deprived for the standard 21 h used in the study. We have now clearly stated this in the Methods section (“Calcium imaging and data analysis” section).

Second, if inhibition of FBI6 neurons represents “rejected choice”, why does bittersweet and not sweet-only taste inhibit FBI6 neurons? Aren’t these two foods in an equilibrium condition, i.e., flies exhibit no preference for either?

We agree that our language did not provide an intuitive interpretation of these results. We would like to clarify that **Fig. 4c-d** depict FBI6 neural responses to only one stimulation with sweet and bittersweet taste in naïve flies, which is different from the conflicting choice assay context. In the choice assay, flies are free to sample both food options repeatedly before consumption. With new animal tracking analyses using recorded videos of flies performing the assay, we have now shown that almost all flies in our choice assay visit both food sectors throughout the assay and transition from one to the other multiple times (**Suppl. Fig. 1g-j**). Even though almost all individual flies sample both options, they consume only one type of food. In the equal-preference condition roughly half of the flies eat only sweet and half eat only bittersweet food. In other

words, it is not that individual hungry flies eat both foods equally, but that half of hungry flies eat only sweet and half eat only bittersweet at the equal preference condition.

From this equal-preference condition, for imaging we selected individual flies that ate sweet only (**Fig. 4c-d**, “choose sweet”), bittersweet only (**Fig. 4c-d**, “choose bittersweet”), or no food (**Fig. 4c-d**, “choose neither”). Since these individual flies ate only one type of food, we were able to map their FB16 neural response to their individual chosen or rejected food option in the preceding behavioral choice assay. For flies that experienced the assay, our results showed that inhibitory FB16 responses corresponded with rejected choice of that individual fly (**Fig. 4c**).

Since flies innately find bittersweet food aversive (Masek, Scott, PNAS 2010, French et al. J Neuro 2015, Lee, Moon and Montell, PNAS 2009, Sellier et al., Chem Senses 2011, French et al., J. Neuroscience 2015) and our results show that FB16 inhibitory activity corresponds with rejected choice, we posit that strong FB16 inhibition by one stimulation with bittersweet taste in naïve food-deprived flies represents rejection of this stimulus. We have now added this point in the Results section (Line 270-273).

We have also added a description of how the preference index was calculated at the beginning of the Results section to aid interpretation of results (new text in Line 52-55), and changed language to clarify what equal-preference means (new text in Line 63-70).

New text in Line 52-55:

Food choice was quantified by scoring the color of the food in each fly’s abdomen at the end of the assay. A preference index was then calculated by subtracting the number of flies that ate bittersweet food from the number of flies that ate sweet food divided by the total number of flies that consumed food in that trial.

New text in Line 63-70:

At 10- and 5-fold sucrose concentration ratios (Fig. 1b, 50 mM vs. 500 mM sucrose + 1 mM quinine, 100 mM sucrose vs. 500 mM + 1 mM quinine at 21 h) almost all flies consumed only a single food option with half of them consuming only sweet and half consuming only bittersweet resulting in equal-preference for sweet and bittersweet at these conditions. The equal-preference point varied with food-deprivation duration, shifting preference from sweet toward bittersweet with increasing food-deprivation duration (Fig. 1b, Supplementary Fig. 1d-f, Supplementary Table 1), suggesting an external taste sensation and internal hunger state equilibrium at the equal-preference concentration ratios.

New text in Line 270-273:

Flies innately find bittersweet food aversive²⁷, and reduce proboscis extension response to and consumption of bittersweet mixtures^{66, 67, 68}. Thus inhibitory taste responses in FB16 to the first encounter with bittersweet food may represent rejection of this option.

Third, if I understand the notion correctly, the authors have assumed that when a fly

chooses one food option, it will choose it again the next time. This assumption may not be correct and requires further investigation. It is known that behavioral expression is probabilistic. For example, flies that choose incorrectly in a memory test display a likelihood of choosing correctly in a retest equal to that of flies who made the correct choice in the initial test (Cervantes-Sandoval & Davis, Curr. Biol. 2012). The authors should validate if flies indeed choose the same option repeatedly in their choice assay.

We thank the reviewer for raising this very interesting point. Here we provide evidence demonstrating that indeed flies choose the same food option repeatedly in our choice assay:

1. We now present animal tracking data at the equal-preference condition showing that ~80% of flies make one and ~69% make more than one transition between the two food sectors (**Suppl. Fig. 1g-i**). Furthermore, these transitions occur throughout the entire duration of the assay (**Suppl. Fig. 1g, i**), and there is no statistically significant difference in the number of transitions made per fly between the first and second half of the assay (**Suppl. Fig. 1j**). This indicates that almost all flies sense both foods multiple times throughout the assay with their leg gustatory receptors.
2. While almost all flies repeatedly sense both foods with their leg gustatory receptors, nevertheless almost all flies consume only one type of food in our assay, as scored by food dye color in their abdomen. This is only possible if flies make the choice to consume the same food repeatedly.

Almost all flies assayed consumed only one type of food, and most flies visited both food options throughout the assay and sensed both food options multiple times with their leg gustatory receptors. Thus, these results demonstrate that flies make the same choice repeatedly in our assay. We have added this discussion in the FBl6 imaging Results section.

New text in Line 284-296:

To test this, we measured FBl6 neural response to sweet and bittersweet tastes in flies that had experienced the equal-preference decision assay. Almost all flies visit both the sweet and bittersweet food sectors (Supplementary Fig. 1h) during the decision task (Supplementary Fig. 1g-i), transitioning from one sector to the other multiple times over the 5 min choice assay. These transits provide flies with sensory information about both food options via gustatory receptors on their legs, since flies are confined to stand on the food surface. The food option that an individual fly consumes is recorded by scoring the color of its abdomen at the end of the assay. Almost all individual flies consume either sweet or bittersweet food only, meaning that under the equal-preference condition approximately 50% of flies ate sweet and the other 50% ate bittersweet food resulting in an equilibrium for the population of flies within a trial. From this equal-preference trial, for imaging we selected individual flies that consumed sweet only, bittersweet only, or no food. Since these individual flies chose only one type of food, we were able to map each fly's subsequent FBl6 neural responses to its chosen and rejected food options.

Fourth, if inhibition of FBl6 neurons represents “rejected choice,” why does silencing FBl6 neurons shift the food preference towards the bittersweet option? The authors’ explanation for this scenario is that it is a default behavioral decision (L322-324), which

is not satisfactory. I suggest the authors further test their model by (1) giving flies a choice between two areas containing identical food and illuminating one area with green light to inhibit FBI6 neurons via GtACR1. Based on their model, the flies should reject the lit area.

We thank the reviewer for this insightful suggestion for additional test of our model. We have performed the suggested experiment by spatially limiting optogenetic FBI6 inhibition to one side of the arena while presenting the same sweet food on both sides (**Fig. 5a**, new). Control flies not expressing GtACR1 avoid food on the lit side of the arena, indicating intrinsic aversion to bright green light used in the assay. This finding is consistent with previously published work showing that flies prefer to eat from dimly lit areas in comparison to brightly lit areas (Rieger et al. J Biol Rhythms). In stark contrast, this intrinsic aversion is abolished in flies expressing GtACR1 in FBI6 neurons (**Fig. 5a**, new), which exhibit equal preference for the lit and dark sides. This indicates that optogenetic inhibition of FBI6 in this “no choice” context induces a preference for the lit side of the arena that counteracts intrinsic green light aversion.

This strong effect of optogenetic inhibition of FBI6 activity in this additional, but different, food-related “no choice” context reinforces FBI6’s key role in food-related decision making. These results again demonstrate that imposing an unnatural inhibitory activity pattern on FBI6 neurons during decision making that is not driven by specific sensory inputs perturbs the decision making process. In these new experiments, optogenetic inhibition of FBI6 is spatially, yet not temporally, confined, unlike our prior experiments with spatially homogenous optogenetic inhibition throughout the food arena (**Fig. 3a**, left panel, **Suppl. Fig. 3b** left panel). Interestingly, these effects are the opposite of what would be expected from a simplistic ON/OFF sensorimotor program in FBI6 that turns feeding on and off on command, independent of the context of sensory environment and internal state at any given moment. Inspired by these new results, we have refined our model to reflect transient and dynamic processing of sensory and internal state information and encoding of choice in these FBI6 neurons.

In light of these new results, we have now adapted a synaptic gating model used in the context of mammalian circuits gating flow of information to the cortex (see below) to explain our results.

New text in Line 317-353:

FB16 neurons are strongly inhibited by a single stimulation with rejected food choice (Fig. 4c-d), this inhibition is context-dependent, and persistent sensory-stimulus independent optogenetic inhibition of FB16 neurons throughout the decision assay shifts food preference to bittersweet (Fig. 3a left panel, Supplementary Fig. 3b left panel). We thus hypothesized that imposing unnatural, temporally and spatially homogenous optogenetic inhibition of FB16 throughout the decision task disrupts value integration and decision making. This imposed activity pattern, that does not take into account the sensory environment and current value estimates of the fly in any given moment, results in the hungry fly defaulting to higher calorie bittersweet food. We tested this model further by optogenetically inhibiting FB16 neurons on only one side of the arena while presenting flies with the same sweet food on both sides. We found that control flies not expressing GtACR1 avoided the lit side of the arena indicating intrinsic aversion of bright green light⁷⁵. In contrast, persistent FB16 optogenetic inhibition mediated by GtACR1 abolished this bright green light avoidance, resulting in equal preference for food on both sides (Fig. 5a, Supplementary Table 1). This indicates that persistent optogenetic inhibition of FB16 induces a positive preference that counteracts the intrinsic aversion to bright green light. While spatially and temporally homogeneous optogenetic inhibition of FB16 neurons during decision making shifted preference to bittersweet (Fig. 3a left panel, Supplementary Fig. 3b left panel), spatially restricted optogenetic inhibition shifted preference to sweet (Fig. 5a). Congruent with context-dependent FB16 neural response to taste stimuli (Fig. 4c-d, Supplementary Fig. 4a-d), optogenetically inhibiting the same FB16 neurons in different behavioral contexts produced different choice outcomes.

These findings are inconsistent with predictions of a simplistic ON/OFF sensorimotor model in which FB16 neurons would turn feeding on and off on command

independent of context. Instead these findings support a model in which the activity of FBl6 neurons is transient, dynamic, and highly state- and context-dependent. We hypothesize that, similar to mammalian nucleus accumbens and other cortical circuits involved in gating the flow of information to cortex for attention and decision-making^{76, 77, 78}, FBl6 neurons are bistable and oscillate between two states that are characterized by different values of membrane potential. According to the synaptic gating model^{76, 77, 78}, in the “down state”, FBl6 neurons would be hyperpolarized (i.e., closed gate) leading to rejection of an option. To reach the “up state”, i.e., a membrane potential just below the neuron’s firing threshold, FBl6 neurons would have to receive input from another neuron. Only when FBl6 neurons are in this “up state” (i.e., open gate) can subsequent inputs trigger action potentials leading to acceptance of an option influenced by these inputs. Similar to mammalian cortical gatekeeper neurons^{76, 77, 78}, FBl6 neurons could act as a gatekeeper that integrates converging inputs from upstream neuropeptidergic neurons using non-linear summation (Fig. 5f). If the activity of FBl6 neurons were forced into an unnatural “down state” using optogenetic inhibition, the integration of inputs to FBl6 neurons would be disrupted. Under such a condition, hardwired innate inputs could be shunted to bypass the gatekeeper circuit motif resulting in unexpected behavioral outcomes that would not take into account adaptive flexible input variables such as the context in which flies choose food options.

(2) Give the flies a food choice by which they will clearly display a preference for one food type over the other. Then activate FBl6 neurons via CsChrimson and establish if this treatment compromises the flies’ decision-making and reduces their preference.

We have performed the suggested experiment under conditions in which flies normally display a clear preference for the sweet food (0.5 M sucrose vs 0.5 M sucrose + 1 mM quinine: **Fig. 1b**, **Suppl. Fig. 1e**). We find no significant difference in food preference between control flies with no FBl6 expression and flies expressing CsChrimson in FBl6 neurons upon shining red light for CsChrimson activation throughout the arena (**Suppl. Fig. 5a**), showing that FBl6 optogenetic activation during the decision assay does not compromise decision making. These results are consistent with expectations, since FBl6 optogenetic activation does not have any effect on food preference at the equal-preference condition in our assay (**Fig. 3a** left panel, **Suppl. Fig. 3b** left panel).

Using the same food options as above (0.5 M sucrose vs 0.5 M sucrose + 1 mM quinine), we performed additional experiments in which we optogenetically activated FBl6 neurons on either only the bittersweet side or only on the sweet side. Again, we found no significant difference in food preference between control flies and flies expressing CsChrimson in FBl6 neurons (**Suppl. Fig. 5b**). These results again suggest that FBl6 optogenetic activation does not affect food preference in the context of our assay.

Fifth, the argument that hungrier flies are also thirstier (L254-256) is unsubstantiated. The authors must directly test this hypothesis by measuring both food and water intake.

We agree that addressing water responses in FBl6 requires further investigation. Since these results are not germane to the key conclusions of this manuscript and merit detailed inquiry beyond the scope of this study, we have removed this section, as suggested by another reviewer.

REVIEWER COMMENTS

Reviewer #1 (Remarks to the Author):

Sareen and colleagues have addressed most of my experimental concerns. The present manuscript convincingly shows that FBL6 neurons labeled by 84c10/c205 integrate neuromodulatory signals from a variety of upstream peptidergic neurons to influence feeding decisions. The current manuscript does a better job of setting the current findings in context. However, several parts of the Results that are primarily focussed on the previous literature or interpretations could probably be trimmed or moved to the Discussion to make the manuscript easier to read (see notes below), and some parts of the manuscript are repetitive. Finally, the conceptual model involving up and down states is highly speculative and not directly supported by the imaging data, which did not detect such states, and therefore belongs in the Discussion, not the Results.

Line 90-102: This does a much better job of citing relevant work but feels out of place in the Results. This section could be moved to the Discussion and the experiments could be introduced simply by starting at the statement “We hypothesized that value...” (line 102)

Line 134-145: This section could also be moved to the Discussion.

Line 149: specific set of mushroom body neurons: do you mean MBONs? KCs?

Line 172-178: can this be stated more simply?

Line 213-215: “The peptidergic receptor for...” This is not a sentence.

Line 305: no-choice assay. I am a little worried that the traces in the supplement are totally flat, unlike those in the naïve, deprived condition. Perhaps worth a note.

Line 317-335: this is an interesting but confusing experiment. Perhaps the presentation and interpretation can be simplified for the reader?

Line 341-347: up and down state model. This is highly speculative and not directly supported by the observation of up and down states in the imaging data. I think this should be moved to the Discussion.

Line 354-379: I appreciate the effort to put the results in context but I found this section fairly confusing. I think a briefer summary in the Discussion of what role the authors think FBL6 neurons play and what remains unknown and should be tested in future experiments would be more helpful for the reader.

Line 395: flight-promoting neurons: I’m not sure where this came from. It seemed out of context.

Reviewer #2 (Remarks to the Author):

The authors have provided detailed and clear responses to my queries and made appropriate changes to the text. I am happy to support publication of this manuscript.

Reviewer #3 (Remarks to the Author):

The authors have made substantial efforts to address my concerns, though I remain unconvinced by some of their responses and disagree with certain data interpretation. However, I have no intention of holding back publication of this work. It is an interesting study, and I am content to see it published in Nature Communications in its current form. Since peer-review should be a continuous process that goes beyond publication of a paper, below I provide some additional comments in the hope that they will be helpful to the authors.

1. In response to my concern about sensory masking in their behavioral assay, the authors argue that the flies' preference for sweet-only and bittersweet foods represents modulated hunger and that sensory masking is hunger-independent. Although the authors have added new data showing that the preference is influenced by starvation, the impact is quite subtle. I have more to say about the effect of starvation on their food choice assay (see my comments below), but here I would like to focus on their argument that sensory masking is independent of hunger. It has been shown that starvation increases sweet and decreases bitter taste sensitivities (Inagaki et al., *Neurons* 2014). Thus, it remains possible that starvation reduces the masking effect of the bitter quinine and shifts the preference towards bittersweet food. In my opinion, the authors have not demonstrated that their choice assay can be used to probe the tradeoff between nutrient value and danger avoidance.

2. The authors show that 21 h of starvation shifts the preference towards bittersweet food only when the sucrose concentration in the sweet-only food is 100 mM but not 50 mM (for W1118 flies). If hunger truly increases flies' evaluation of caloric content in the food, how can the preference shift only occur when 100 mM, but not 50 mM, sucrose is used? Moreover, most of the experiments in this study were done in the W1118 background with 50 mM sucrose in the sweet-only food. Since starvation does not affect food preference under this condition, the authors' later interpretation that LK neuron inhibition (resulting in preference shifting towards bittersweet food) reflects an increase in perceived hunger is not appropriate (L127).

3. The authors have not explicitly addressed the potential conflicts between their results and other published works about the roles of the neuropeptides. Instead, they use a general discussion to argue that the functions of neuropeptides can be context-dependent. I agree that neuropeptide function can be context-dependent. However, I think more specific comparisons of their results with other published works would help highlight these new findings with respect to current knowledge.

4. If inhibition of FBI6 neurons represents a rejected choice, why do the naïve deprived flies show a strong inhibited FBI6 response to the bittersweet food, yet ultimately, half of these flies consume this food exclusively in the choice assay? It would seem that FBI6 activity cannot predict behavioral outcome, and the activity patterns the authors observed in the last three conditions of Fig. 4c might be consequences of repeated decision-making rather than part of the decision-making process.

5. How can the gating model in Fig. 5f explain that inhibition of FBI6 on one side of the choice chamber drives attraction to that side (Fig. 5a)?

6. Line 148, it should be optogenetic silencing for PPL1-a3.

7. Line 149-157, why not silencing the MB neurons? The conclusion that dopaminergic neurons modulate food choice via non-MB regions appears to be flawed when it is only based on optogenetic activation but not silencing of the MB neurons.

Reviewer #1 (Remarks to the Author):

Sareen and colleagues have addressed most of my experimental concerns. The present manuscript convincingly shows that FBL6 neurons labeled by 84c10/c205 integrate neuromodulatory signals from a variety of upstream peptidergic neurons to influence feeding decisions. The current manuscript does a better job of setting the current findings in context. However, several parts of the Results that are primarily focussed on the previous literature or interpretations could probably be trimmed or moved to the Discussion to make the manuscript easier to read (see notes below), and some parts of the manuscript are repetitive. Finally, the conceptual model involving up and down states is highly speculative and not directly supported by the imaging data, which did not detect such states, and therefore belongs in the Discussion, not the Results.

We thank Reviewer #1 for their helpful suggestions. As suggested, we have moved parts of the Results to the Discussion and clarified language where requested. Please see our point-by-point response below.

Line 90-102: This does a much better job of citing relevant work but feels out of place in the Results. This section could be moved to the Discussion and the experiments could be introduced simply by starting at the statement “We hypothesized that value...” (line 102)

As suggested, this text has been moved to the Discussion (Line 395 – 408).

Line 134-145: This section could also be moved to the Discussion.

Text moved to the Discussion (Line 426 – 436).

Line 149: specific set of mushroom body neurons: do you mean MBONs? KCs?

We have added the specific neuron name, KC, in the text now. New text in Line 151 – 153:

Optogenetic activation of various subsets of mushroom body Kenyon cells (KC), a brain region that controls higher-order behaviors and receives projections from the identified dopaminergic neurons, did not affect preference (Fig. 2a).

Line 172-178: can this be stated more simply?

We have rephrased our language for clarity. New text in Line 177 – 180:

By “encoding” of hunger state we mean the estimation of hunger level by specific neurons. By “integration” we mean combining of hunger state with other internal state and external sensory information.

Line 213-215: “The peptidergic receptor for...” This is not a sentence.

We have now rephrased Line 213 – 215. New text in Line 216 – 218:

The receptors for AstA, DH44, and Lk are GPCRs that potentially couple to distinct G proteins in FBl6 neurons, which in turn interact with different effector molecules to produce distinct downstream cellular responses.

Line 305: no-choice assay. I am a little worried that the traces in the supplement are totally flat, unlike those in the naïve, deprived condition. Perhaps worth a note.

While it is perhaps arguable surprising to see such a difference, these flies had a very different prior experience than naïve flies, which could underlie the observed differences in FBl6 activity. The flies in these new experiments explored and consumed food of only one type before imaging. We agree that these findings are worth exploring further in the future.

Line 317-335: this is an interesting but confusing experiment. Perhaps the presentation and interpretation can be simplified for the reader?

We have rephrased our language for clarity. New text in Line 320 – 339:

FBl6 neurons are strongly inhibited by a single stimulation with rejected food choice (Fig. 4c-d), this inhibition is context-dependent, and persistent sensory-stimulus independent optogenetic inhibition of FBl6 neurons shifts food preference to bittersweet (Fig. 3a left panel, Supplementary Fig. 3b left panel). We thus hypothesized that imposing unnatural, temporally and spatially homogenous optogenetic inhibition of FBl6 throughout the decision task disrupts value integration and decision making. This imposed activity pattern, that does not take into account the sensory environment and current value estimates of the fly in any given moment, results in the hungry fly defaulting to higher calorie bittersweet food. To test this further, we optogenetically inhibited FBl6 neurons on only one side of the arena while presenting flies with the same sweet food on both sides. We found that control flies not expressing GtACR1 avoided the lit side of the arena indicating intrinsic avoidance of bright green light⁷⁵. In contrast, persistent FBl6 optogenetic inhibition abolished this bright green light avoidance, resulting in equal preference for food on both sides (Fig. 5a, Supplementary Table 1). While spatially and temporally homogeneous optogenetic inhibition of FBl6 neurons shifted preference to bittersweet (Fig. 3a left panel, Supplementary Fig. 3b left panel), spatially restricted optogenetic inhibition shifted preference to sweet (Fig. 5a). Congruent with context-dependent FBl6 neural response to taste stimuli (Fig. 4c-d, Supplementary Fig. 4a-d), optogenetically inhibiting the same FBl6 neurons in different behavioral contexts produced different choice outcomes.

Line 341-347: up and down state model. This is highly speculative and not directly supported by the observation of up and down states in the imaging data. I think this should be moved to the Discussion.

As suggested, we have moved the model (previous Line 314 – 347) that is supported in part by our imaging data to the Discussion (Line 452 – 459).

Line 354-379: I appreciate the effort to put the results in context but I found this section fairly confusing. I think a briefer summary in the Discussion of what role the authors

think FBL6 neurons play and what remains unknown and should be tested in future experiments would be more helpful for the reader.

Lines 354 – 356 have been deleted. Text in Line 356 – 361 was added at the request of another reviewer and has been moved to the Discussion (Line 414 – 418). Text in Line 371 – 379 has been deleted.

Line 395: flight-promoting neurons: I'm not sure where this came from. It seemed out of context.

This text was added at the request of another reviewer.

Reviewer #2 (Remarks to the Author):

The authors have provided detailed and clear responses to my queries and made appropriate changes to the text. I am happy to support publication of this manuscript.

We thank Reviewer #2 for their comments and suggestions that helped improve this manuscript.

Reviewer #3 (Remarks to the Author):

The authors have made substantial efforts to address my concerns, though I remain unconvinced by some of their responses and disagree with certain data interpretation. However, I have no intention of holding back publication of this work. It is an interesting study, and I am content to see it published in Nature Communications in its current form. Since peer-review should be a continuous process that goes beyond publication of a paper, below I provide some additional comments in the hope that they will be helpful to the authors.

We thank Reviewer #3 for their affirmation of the interest in our study and that it is now appropriate for publication in its current form. We agree that the additional comments below regarding theoretical interpretation of data will be useful not only to us as we further explore these phenomena, but also to the readers of the published paper in the event that the peer review dialogue is published along with the paper, which we wholeheartedly endorse. Please find our point-by-point response below.

1. In response to my concern about sensory masking in their behavioral assay, the authors argue that the flies' preference for sweet-only and bittersweet foods represents modulated hunger and that sensory masking is hunger-independent. Although the authors have added new data showing that the preference is influenced by starvation, the impact is quite subtle. I have more to say about the effect of starvation on their food choice assay (see my comments below), but here I would like to focus on their argument that sensory masking is independent of hunger. It has been shown that starvation increases sweet and decreases bitter taste sensitivities (Inagaki et al., Neurons 2014). Thus, it remains possible that starvation reduces the masking effect of the bitter quinine and shifts the preference towards bittersweet food. In my opinion, the authors have not demonstrated that their choice assay can be used to probe the tradeoff between nutrient value and danger avoidance.

We agree with the reviewer that sensory masking can be hunger dependent and in line with this view we never made the argument that sensory masking is independent of hunger. We also agree that since hunger has been shown to change sweet and bitter sensitivities, it is almost certain that starvation plays a role in shifting the balance between sweet and bittersweet at the sensory level in our assay. In fact, in our current manuscript we are specifically focused on how hunger shifts this balance between sweet and bittersweet conflicting sensory food cues. We believe the tradeoff happens at multiple levels and not just the sensory level, which is also in line with the discussion by Inagaki *et al.*, Neuron 2014. While our assay is not specifically designed to probe the tradeoff between nutrient value and danger avoidance, this was a straightforward interpretation of the tradeoff observed between sweet and bittersweet food options in our assay since the changes in sweet and bitter sensitivities due to starvation are thought to be a mechanism to maintain energy balance with potential toxin avoidance (Inagaki *et al.*, Neuron 2014).

2. The authors show that 21 h of starvation shifts the preference towards bittersweet food only when the sucrose concentration in the sweet-only food is 100 mM but not 50 mM (for W1118 flies). If hunger truly increases flies' evaluation of caloric content in the food, how can the preference shift only occur when 100 mM, but not 50 mM, sucrose is used? Moreover, most of the experiments in this study were done in the W1118 background with 50 mM sucrose in the sweet-only food. Since starvation does not affect food preference under this condition, the authors' later interpretation that LK neuron inhibition (resulting in preference shifting towards bittersweet food) reflects an increase in perceived hunger is not appropriate (L127).

Food deprivation shifts the entire preference index curve in our assay (Fig. 1b, Suppl. Fig. 1e), even though we have performed statistics only on the equilibrium condition. Additionally, transgenic flies that are generated in *w1118* genetic background often have behavioral sensitivities different from *w1118* only strain. Therefore, the limited dynamic range of preference index for *w1118* flies does not preclude flies with different transgenes inserted into *w1118* background from having a larger behavioral range at 50 mM condition, just like CS wild-type flies. Since hunger does shift the entire preference index curve at different sweet and bittersweet concentrations, in our opinion, the interpretation about Lk neuron inhibition reflecting an increase in perceived hunger is appropriate because its opposite manipulation, i.e., Lk neuron activation, has the opposite effect of abolishing food consumption in hungry flies.

3. The authors have not explicitly addressed the potential conflicts between their results and other published works about the roles of the neuropeptides. Instead, they use a general discussion to argue that the functions of neuropeptides can be context-dependent. I agree that neuropeptide function can be context-dependent. However, I think more specific comparisons of their results with other published works would help highlight these new findings with respect to current knowledge.

In our previous response, we addressed all of the results that this Reviewer considered conflicting with published works. We also explained how and why we do not see our results in conflict with other published works. We appreciate this Reviewer's interest in these comparisons but we believe that a literature review of each neuropeptide may be outside the scope of the current manuscript's Discussion and may take away from our main focus on the role of FB16 neurons in this manuscript.

4. If inhibition of FBI6 neurons represents a rejected choice, why do the naïve deprived flies show a strong inhibited FBI6 response to the bittersweet food, yet ultimately, half of these flies consume this food exclusively in the choice assay? It would seem that FBI6 activity cannot predict behavioral outcome, and the activity patterns the authors observed in the last three conditions of Fig. 4c might be consequences of repeated decision-making rather than part of the decision-making process.

The neural activity that we observe while imaging from naïve food deprived flies reports only the first encounter of the bittersweet food taste, which appears to show that all naïve flies will reject bittersweet at first encounter. This is in line with published literature, as pointed out in our previous point-by-point response. We have argued that FBI6 neural activity is dynamic and changes with a flies changing experience. Decisions are constantly updated as an animal experiences and interacts with its environment, which is part of the decision making process. We expect that as decisions are updated and changed, so does the FBI6 neural activity reflecting these decisions.

5. How can the gating model in Fig. 5f explain that inhibition of FBI6 on one side of the choice chamber drives attraction to that side (Fig. 5a)?

According to the gating model a context-dependent hyperpolarized “down state” will result in rejection of an option. However, in the Fig. 5f we imposed an unnatural activity pattern on FBI6 neurons that likely overrides the context-dependent gating activity of FBI6 neurons. This results in disruption of decision making and unexpected results.

6. Line 148, it should be optogenetic silencing for PPL1-a3.

Thank you for pointing this out. We have corrected the optogenetic manipulation condition for PPL1-a3 in the main text. New text in Line 149 – 151:

Specifically, optogenetic activation of PPL1- γ 2 α '1 and PAM-a1, while optogenetic inhibition of PPL1-a3 subsets shifted preference towards bittersweet (Fig. 2a).

7. Line 149-157, why not silencing the MB neurons? The conclusion that dopaminergic neurons modulate food choice via non-MB regions appears to be flawed when it is only based on optogenetic activation but not silencing of the MB neurons.

We did not test KC/MB optogenetic inhibition since our activation screen did not yield any changes in preference compared to control flies. Studies that have shown mushroom body KCs to be involved in other types of decision making have found that KC activation has an effect on the specific type of decision making that they assay (Lewis *et al.*, Current Biology 2015). We did conduct silencing experiments with TNTe targeting all MB lobes (201Y-GAL4 and MB010B-GAL4 > UAS-TNTe) and found no shift in preference away from equal preference. These data are available, if requested. However, we do agree with the reviewer that acute KC inhibition could still be involved in our decision making behavior and would be important to test in future studies.